# Enhanced Apoptosis and Loss of Cell Viability in Melanoma Cells by Combined Inhibition of ERK and Mcl-1 Is Related to Loss of Mitochondrial Membrane Potential, Caspase Activation and Upregulation of Proapoptotic Bcl-2 Proteins

**DOI:** 10.3390/ijms24054961

**Published:** 2023-03-04

**Authors:** Zhe Peng, Bernhard Gillissen, Antje Richter, Tobias Sinnberg, Max S. Schlaak, Jürgen Eberle

**Affiliations:** 1Skin Cancer Centre Charité, Department of Dermatology, Venereology and Allergology, Charité—Universitätsmedizin Berlin, Charitéplatz 1, 10117 Berlin, Germany; 2Clinical Medicine, University of South China, Hengyang 421001, China; 3Department of Hematology, Oncology, and Tumor Immunology, Charité—Universitätsmedizin Berlin, 13125 Berlin, Germany; 4Division of Dermatooncology, Department of Dermatology, University Tübingen, 72076 Tübingen, Germany

**Keywords:** melanoma, apoptosis, Bcl-2 proteins, Mcl-1, Bim, Puma, Noxa

## Abstract

Targeting of MAP kinase pathways by BRAF inhibitors has evolved as a key therapy for BRAF-mutated melanoma. However, it cannot be applied for BRAF-WT melanoma, and also, in BRAF-mutated melanoma, tumor relapse often follows after an initial phase of tumor regression. Inhibition of MAP kinase pathways downstream at ERK1/2, or inhibitors of antiapoptotic Bcl-2 proteins, such as Mcl-1, may serve as alternative strategies. As shown here, the BRAF inhibitor vemurafenib and the ERK inhibitor SCH772984 showed only limited efficacy in melanoma cell lines, when applied alone. However, in combination with the Mcl-1 inhibitor S63845, the effects of vemurafenib were strongly enhanced in BRAF-mutated cell lines, and the effects of SCH772984 were enhanced in both BRAF-mutated and BRAF-WT cells. This resulted in up to 90% loss of cell viability and cell proliferation, as well as in induction of apoptosis in up to 60% of cells. The combination of SCH772984/S63845 resulted in caspase activation, processing of poly (ADP-ribose) polymerase (PARP), phosphorylation of histone H2AX, loss of mitochondrial membrane potential, and cytochrome c release. Proving the critical role of caspases, a pan-caspase inhibitor suppressed apoptosis induction, as well as loss of cell viability. As concerning Bcl-2 family proteins, SCH772984 enhanced expression of the proapoptotic Bim and Puma, as well as decreased phosphorylation of Bad. The combination finally resulted in downregulation of antiapoptotic Bcl-2 and enhanced expression of the proapoptotic Noxa. In conclusion, combined inhibition of ERK and Mcl-1 revealed an impressive efficacy both in BRAF-mutated and WT melanoma cells, and may thus represent a new strategy for overcoming drug resistance.

## 1. Introduction

Melanoma incidence has steadily increased since the 1960s and accounts for 90% of deaths from skin cancer [1,2]. It is one of the most severely mutated malignancies, with about 50% mutations in the mitogen-activated protein kinase (MAPK) BRAF, mainly BRAF V600E mutations [3,4]. After a long period without efficient therapy, two new therapeutic strategies have now significantly improved the prognosis of metastatic melanoma. Firstly, selective inhibitors of the MAP kinases BRAF and MEK were established for targeted therapy [5], and secondly, anti-CTLA4 and anti-PD1 antibodies were established for stimulation of an antitumor immune response [6,7].

Restricting the use of BRAF inhibitors however, they may only be used for BRAF-mutated melanoma (about 50%), and the response to therapy is usually limited, due to the development of resistance over the course of treatment [8,9]. In addition, treatment-related adverse effects may decrease a patient’s quality of life [10]. On the other hand, immunotherapies (anti-CTLA4, anti-PD1, or the combination of both) can produce long-term therapeutic successes, but are also effective in only a portion of patients (30–50%). In addition, the side effect rate of these therapies can be considerable, as disruption of a functional immune checkpoint may lead to an uncontrolled immune response against the skin, gastrointestinal system, liver, lungs, or other organs [11,12]. Thus, despite the significant progress made in melanoma therapy, still, large therapeutic gaps remain, and many melanoma patients cannot be treated sufficiently. The development of new therapeutic strategies, focusing on higher efficacy and overcoming resistance, is still challenging.

Most anticancer therapies target abnormal tumor cell proliferation and/or induction of apoptosis [13]. Characteristic hallmarks of apoptosis are the early phosphatidylserine flip-flop in the cytoplasma membrane, as well as the late step of double-strand breaks and DNA fragmentation. For induction of apoptosis, extrinsic and intrinsic pathways have been described. Thus, extrinsic pathways are triggered by binding of death ligands, such as tumor necrosis factor-α (TNF-α), CD95 (Fas) ligand, or TNF-related apoptosis-inducing ligand (TRAIL), to their respective death receptors, which results in the formation of a death-inducing signaling complex, where initiator caspase-8 and/or -10 are activated. In a caspase cascade, downstream effector caspases such as caspase-3 and caspase-7 are cleaved, and thus activated by initiator caspases. They irreversibly ensure the execution of apoptosis by cleaving a large number of death substrates, e.g., poly (ADP-ribose) polymerase (PARP) [14,15].

On the other hand, intrinsic apoptosis pathways can be induced by cellular stress conditions, such as by chemotherapy or DNA damage. The activation of mitochondria in intrinsic proapoptotic pathways results in the release of mitochondrial proapoptotic factors, such as cytochrome c, and loss of mitochondrial membrane potential [16,17]. Furthermore, evidence is increasing for additional roles of reactive oxygen species (ROS), which were shown, in melanoma cells, to be involved in apoptosis induction by different antitumor strategies [18,19]. 

Intrinsic apoptosis pathways are crucially regulated by the family of pro- and antiapoptotic Bcl-2 proteins, which also regulate each other through hetero-dimerization. Thus, the multidomain, proapoptotic proteins Bax and Bak mediate mitochondrial permeability and are antagonized by antiapoptotic Bcl-2 proteins (e.g., Bcl-2, Bcl-x_L_, Bcl-w, and Mcl-1). Finally, proapoptotic BH3-only proteins (e.g., Bim, Bad, Puma, and Noxa) function as triggers in apoptosis control through binding and antagonizing antiapoptotic Bcl-2 proteins or directly activating Bax [16,17,20]. Cancer cells often require high activity of antiapoptotic Bcl-2 proteins for cell survival and inhibition of apoptosis. Furthermore, extrinsic apoptotic pathways are often weak and need to be reinforced by mitochondrial pathways, also shown in melanoma cells [15]. 

As concerning MAPK pathways, their activation, via Ras-BRAF-MEK-ERK, represents an important step in melanoma [21], driving enhanced cell proliferation, inhibition of apoptosis, invasion, and metastasis [22,23]. Besides the approved inhibitors for mutated BRAF and MEK, inhibitors of the downstream ERK come into consideration as alternative therapeutic strategies. Thus, SCH772984 has been developed as an ATP-competitive inhibitor of ERK1/2, which inhibits kinase activity and prevents phosphorylation of ERK by MEK [24]. 

In addition, inhibitors of antiapoptotic Bcl-2 proteins may serve as alternative therapeutic strategies. As for Mcl-1, its oncogenic activities have drawn particular attention in recent years, due to frequent amplifications of the Mcl-1 gene in about 40% of cancer cells of different origin [25], and due to an often high Mcl-1 expression, which was correlated to tumor progression and drug resistance [26]. The small molecule S63845 specifically binds to the BH3-binding domain of Mcl-1, and thus can kill Mcl-1-dependent cancer cells by activating Bax/Bak-dependent apoptosis pathways [27]. 

With the goal of improving the therapeutic effectiveness, and overcoming drug resistance in melanoma, we combined vemurafenib, as well as SCH772984 with the Mcl-1 inhibitor S63845, and report here strongly enhanced effects on cell viability, apoptosis, and on proapoptotic pathways.

## 2. Results

### 2.1. Strongly Decreased Cell Viability by Combination of MAPK and Mcl-1 Inhibitors

For identification of suitable strategies to target melanoma, effects of the BRAF inhibitor vemurafenib (10, 20, 30 µM), the pERK inhibitor SCH772984 (0.1, 1, 10 µM), and the Mcl-1 inhibitor S63845 (0.25, 0.5, 1 µM) were investigated in BRAF-mutated melanoma cell lines (A-375 and Mel-HO), as well as in BRAF-WT cell lines (MeWo and SK-Mel-23). Single treatments with these inhibitors resulted in only rather limited effects in terms of reduction in cell viability, as determined by calcein staining and flow cytometry. Thus at 48 h, the approved BRAF inhibitor vemurafenib, reduced cell viability in BRAF-mutated cell lines down to 64% (A-375) and to 66% (Mel-HO), and even less response was seen in BRAF-WT cells (MeWo, 85%; SK-Mel-23, 76%; controls: 100%). The effects of SCH772984 were comparable in BRAF-mutated and BRAF-WT cells (A-375, 68%; Mel-HO, 60%; MeWo, 65%; SK-Mel-23; 69%; 10 µM). Also the effects of S63845 alone on cell viability were limited (A-375, 88%; Mel-HO, 95%; MeWo, 65%; SK-Mel-23, 90%; 48 h; 1 µM; Figure 1).

In clear contrast, the effects of MAPK inhibitors were strongly enhanced when combined with S63845. Thus, the combination of vemurafenib (30 µM) and S63845 (1 µM) almost completely abolished cell viability in A-375 and Mel-HO (6%, 7%), and even in BRAF-WT cell lines, cell viability was further decreased to 47% (MeWo) and 32% (SK-Mel-23), respectively. Demonstrating a general responsiveness of melanoma cells to ERK inhibition, both BRAF-mutated and BRAF-WT cells responded with strong loss of cell viability to combination treatments with SCH772984/S63845 (A-375, 8%; Mel-HO, 11%; MeWo, 17%; SK-Mel-23, 7%; 10 µM SCH772984/1 µM S63845; 48 h; Figure 1). Largely comparable effects on cell viability were also determined at 24 h (Appendix A).

### 2.2. Loss of Cell Proliferation in Parallel with Decreased Cell Viability

Largely parallel findings were obtained at the level of cell proliferation, as determined in A-375 and MeWo by WST-1 assay at 24 and 48 h. While MeWo cells were less responsive to vemurafenib alone (67%), as compared to A-375 (43%; vemurafenib, 30 µM, 48 h), MeWo revealed higher sensitivity to SCH772984 (17%) than A-375 (52%; SCH772984, 10 µM, 48 h; Figure 2).

The antiproliferative effects of vemurafenib and SCH772984 were enhanced in both cell lines when combined with S63845. Thus, in BRAF-mutated A-375, cell proliferation rates, already decreased in response to 30 µM vemurafenib at 24/48 h (58/43%), further dropped in response to combination treatment with 1 µM S63845 (15/8%). Also in the BRAF-WT cell line MeWo, cell proliferation rates at 24/48 h in response to 30 µM vemurafenib (76/67%) were somewhat further reduced in response to the combination with 1 µM S63845 (58/31%). As for SCH772984, strong combination effects were seen in both cell lines. Thus, cell proliferation rates that had been decreased in A-375 to 81/52% by 10 µM SCH772984 at 24/48 h, further dropped to 16/13% in response to 10 µM SCH772984/1 µM S63845. Similarly, in MeWo, cell proliferation rates (64/52%; 1 µM SCH772984) dropped to 32/20% (1 µM SCH772984/1 µM S63845; Figure 2).

### 2.3. Decreased Cell Viability Correlates with Induction of Apoptosis 

For unravelling the causes of decreased viability and cell proliferation, induction of apoptosis was determined at 24 h and at 48 h by cell cycle analysis in the four cell lines. In response to vemurafenib alone (30 µM), apoptosis was induced in BRAF-mutated cells at 48 h to 36% (A-375) and 14% (Mel-HO), respectively, while BRAF-WT cell lines were largely resistant (7/5%; Figure 3). Only little effects on apoptosis induction was seen in response to SCH772984 single treatment in each of the four cell lines (6%, 11%, 8%, 4%), while some apoptosis was induced in response to S63845 alone in A-375 and MeWo (14/24%; Figure 3). 

Apoptosis was strongly enhanced in combinations of MAPK inhibitors and S63845. Thus, in response to (30 µM vemurafenib/1 µM S63845), apoptosis at 48 h increased to 60% (A-375, Mel-HO), whereas BRAF-WT cells were less responsive, reaching at maximum 17% (SK-Mel-23) and 35% (MeWo). In particular, all four cell lines were strongly responsive to combinations of S63845/SCH772984, resulting in apoptotic rates of up to 56% (A-375), 55% (Mel-HO), 64% (MeWo), and 44% (SK-Mel-23; Figure 3). Comparable effects were seen at 24 h (Appendix A).

Induction of cell death in response to 1 µM SCH772984/1 µM S63845 was further substantiated in A-375 and MeWo at 24 h and 48 h by Annexin V-FITC/PI staining (AnnV/PI). Early apoptotic cells were identified as AnnV(+)/PI(−), and late apoptotic or necrotic cells were identified as AnnV(+)/PI(+) (Figure 4). In response to single treatments, cell death induction (all AnnV(+) cells) was only slightly induced. Thus, in response to S63845, AnnV(+) cells were at 21%/28% in MeWo, and at 5%/7% in A-375, at 24/48 h. The strongest effects were obtained by combination treatments, resulting in 65% in A-375 and 71% in MeWo at 48 h. Early apoptotic cells, AnnV(+)/PI(−), were more represented in MeWo, while AnnV(+)/PI(+) cells were more frequent in A-375 after combination treatment. The increase in both cell death fractions by combination treatment was highly significant, as compared to the single treatments as well as compared to control cells (*p* < 0.01; Figure 4). 

To identify synergism, drug combination indices were calculated by the program SyngeryFinder 3.0 [28]. The combination SCH772984/S63845 was determined to be synergistic in all four cell lines (δ scores > 10). Thus, for decreased cell viability, values of 48, 38, 30, and 55 were calculated for A-375, Mel-HO, MeWo, and SK-Mel-23, respectively (Appendix A). In agreement, the δ scores for induced apoptosis ranged between 21 and 30 (Appendix A). On the other hand, the combination vemurafenib/S63845 appeared to be particularly strongly synergistic in BRAF-mutated cell lines, with δ scores for cell viability of 54 (A-375) and 50 (Mel-HO), as well as δ scores for apoptosis of 15 (A-375) and 30 (Mel-HO). In contrast, the effects of vemurafenib/S63845 were less pronounced in BRAF-WT cell lines (cell viability: 9 (MeWo) and 32 (SK-Mel-23); apoptosis: 5 (MeWo) and 4 (SK-Mel-23); Appendix A). The δ scores for decreased cell proliferation rates were in agreement (Appendix A).

For obtaining some understanding of the response of normal cell populations, we investigated the effects of SCH772984 (1 µM) and S63845 (1 µM) in three cultures of normal human fibroblasts. These cells showed no significant effects to the single treatments and responded much less to the combination of SCH772984/S63845, as compared to melanoma cells: reduced cell viability was at 82% +/− 7% and induction of apoptosis at 11% +/− 2% (mean values and SDs of three experiments; Appendix A).

### 2.4. Loss of mMP and Production of ROS

Loss of mitochondrial membrane potential (mMP) represents a characteristic step in intrinsic apoptosis pathways. It was determined, by TMRM^+^ staining and flow cytometry, in A-375 and MeWo in response to SCH772984 (1 µM, 10 µM), vemurafenib (30 µM), S63845 (1 µM), and the combinations. As for vemurafenib/S63845, strong loss of mMP was seen in A-375 at 24 h (59%), whereas MeWo was almost not responsive. In contrast, a significant loss of mMP was seen in both cell lines in response to the combination SCH772984/S63845 (A-375, 33/79%; MeWo, 66/38%; for 1/10 µM SCH772984 and 1 µM S63845 (Figure 5a)).

Loss of mMP in cells treated with SCH772984 (1 µM)/S63845 (1 µM) was further visualized by JC-1/Hoechst-33342 double staining, in a time kinetic analysis (4, 8, 16, 24, and 48 h). While cell nuclei are stained blue with Hoechst-33342, the cationic dye JC-1 accumulates in mitochondria of viable cells, where it forms red fluorescent aggregates. Upon loss of mMP, JC-1 locates to the cytosol, and fluorescence shifts from red to green. A-375 and MeWo cells showed clear signs of lost mMP (green cells) at 24 h, which further increased at 48 h. In contrast, no significant loss of mMP was determined for single treatments or at earlier times (Figure 6).

Production of reactive oxygen species (ROS) may accompany the activation of intrinsic apoptosis pathways. Indeed, ROS levels were enhanced at 24 h after sole MAPK inhibition in A-375 to 27% (10 µM SCH772984) and 39% (30 µM vemurafenib), as well as in MeWo to 40% (10 µM SCH772984) and 36% (30 µM vemurafenib), as determined by H_2_DCF staining and flow cytometry. The ROS levels at 24 h were further enhanced by the combination 10 µM SCH772984/1 µM S63845 (A-375, up to 67%; MeWo, up to 58%; Figure 5b).

### 2.5. Significant Role of Caspase Activation 

Further investigation of the activated pathways in response to SCH772984/S63845 in A-375 and in MeWo revealed strong suppression of ERK phosphorylation by SCH772984 (1 µM) at 4 h (median values for MeWo/A-375: 28%/14%) and at 24 h (13%/10%, as compared to controls; Figure 7A,B). 

Suggesting a critical contribution of proapoptotic caspase cascades, activated cleavage products of the major effector caspase-3 (15/17 kD), were found in both cell lines after combination treatment, as shown by two antibodies against cleaved caspase-3 and total caspase-3, respectively (Figure 7B,C). Caspase-8, the initiator caspase of the extrinsic pathway (cleavage product: 41 kD), was upregulated in A-375 in response to combination treatment, while caspase-9, the initiator caspase of the intrinsic pathway (cleavage products: 37/20 kD), was upregulated in both cell lines (Figure 7B,C). 

Further indicating caspase activity, PARP (poly (ADP-ribose) polymerase, 89 kDa, 24 kDa) was cleaved in response to combination treatments, whereas it was not induced in controls or SCH772984-treated cells. PARP cleavage in response to S63845 was less pronounced, and the 89 kDa product was 2.5-fold stronger expressed in MeWo and 13-fold stronger in A-375 after combination treatment vs. S63845 alone (Figure 7C). Caspase activity was also determined by quantitative assays, measuring caspase-8, caspase-9, and caspase-3/7 activity (FLICA Apoptosis Detection kits). These showed consistent caspase activation in 37% of MeWo and in 28% of A-375 cells (Appendix A).

Phosphorylation of histone H2AX (γH2AX) is characteristically induced in cells with DNA double strand breaks and fragmented DNA [29]. Clearly underlining the previous findings of DNA fragmentation by sub-G1 analyses (Figure 3), γH2AX was strongly induced after combination treatment, whereas not detected in controls and SCH772984-treated cells. Some γH2AX was also seen in MeWo in response to S63845 alone, which was increased by 7.3-fold upon combination treatment (Figure 7C).

Supporting the activation of proapoptotic mitochondrial pathways, we found strongly induced mitochondrial release of cytochrome c into the cytosolic cellular compartment in response to combination treatment. No cytochrome c was detected in cytosolic extracts of controls and SCH772984-treated cells. Some cytochrome c was also detected in the cytosol after S63845 treatment, which was enhanced in the combinations by 2.7-fold (MeWo) and 7.5-fold (A-375; Figure 7D).

Thus, several steps in the apoptosis signaling cascades were partly activated already by S63845 alone, i.e., caspase-9, PARP, pH2AX, and cytochrome c. This was, however, less pronounced, and full activation appeared only after combination treatment in both cell lines.

To prove the significance of caspase activation for the antitumor effects of SCH772984/S63845, the pan-caspase inhibitor QVD-Oph (QVD) was applied. Caspase inhibition by QVD completely abolished apoptosis induction at 24 h by SCH772984/S63845 (1 µM) in A-375 (40% → 3%) and in MeWo (39% → 3%; Figure 8a), and it largely restored cell viability at 24 h (A-375, 28% → 62%; MeWo, 30% → 80%; Figure 8b).

As concerning signaling via mMP and ROS, the two cell lines showed a different response to QVD. While QVD strongly diminished loss of mMP in MeWo (66% → 31%; Figure 8c), it abolished ROS induction in A-375 (34% → 12%; Figure 8d), suggesting a different role for these two effects in intrinsic apoptosis pathways in both cell lines. We also aimed to investigate the possible roles of ROS by using the antioxidants N-acetylcysteine (NAC, 1 mM) or α-tocopherol (vitamin E, 1 mM). However, neither NAC nor tocopherol could significantly reduce ROS production by SCH772984/S63845, while ROS production by an indirubin derivative (DKP-073, 10 µM [18]), used as control, was completely abrogated by NAC (Appendix A). In consequence, antioxidants had no significant effect on cell viability (Appendix A) or apoptosis (Appendix A).

### 2.6. Upregulation of Proapoptotic BH3-Only Proteins

Intrinsic apoptosis pathways are critically controlled by the family of pro- and antiapoptotic Bcl-2 proteins, and their up- and downregulation may play critical roles in antitumor therapy. We investigated, by Western blotting, the expression of antiapoptotic Mcl-1 and Bcl-2, as well as of proapoptotic Bax, Bad, Bim, Puma, and Noxa in MeWo and A-375 at 24 h, in response to treatment with SCH772984 (1 µM) and S63845 (1 µM). Mcl-1 protein expression was found to be upregulated in both cell lines by S63845, which may be understood as a cellular response to compensate for the inhibition of Mcl-1 activity by S63845, as also reported previously [27]. Median induction factors were calculated on the basis of densitometric, semiquantitative analyses, and normalization with the GAPDH signals from two independent series of protein extracts and Western blots. Thus, median induction factors for Mcl-1 by S63845 were at 2.2-fold (MeWo) and 4.0-fold (A-375). Indicative of a suitable combination effect, this upregulation was completely abolished by the combination with SCH772984 (Figure 9), also suggesting that Mcl-1 upregulation may depend on MAPK activation. Bcl-2 was slightly downregulated by SCH772984 (MeWo, 61%; A-375, 51%), whereas no significant changes were obtained for proapoptotic Bax at 24 h. 

As concerning proapoptotic BH3-only proteins, two protein bands were obtained with the Bim antibody (23, 24 kD), of which 24 kD corresponds in size to Bim_EL_. Clearly, both bands were upregulated by SCH772984 in the two cell lines at 24 h (MeWo, 1.6-fold; A-375, 2.8-fold). Also Puma was upregulated by SCH772984 (MeWo, 4.5-fold; A-375, 2.6-fold), reflecting the roles of MAPK pathways for their expression. In addition, Noxa was upregulated, in the course of combination treatment, in both cell lines (MeWo, 6.6-fold; A-375, 11-fold; Figure 9). Finally, we found a decrease in phosphorylated Bad (pBad) by SCH772984 in A-375 at 4 h (SCH772984, 50%; SCH772984/S63845, 38%), while total Bad remained unchanged. The phosphorylated pBad represents the inactive form of Bad, and thus may indicate Bad activation. These data suggest that BH3-only proteins are critically involved in the antitumor effects of combination treatment by SCH772984/S63845.

## 3. Discussion

While there was no suitable therapy for metastasized melanoma available until about a decade ago, prognoses of melanoma patients have substantially improved in recent years, due to the identification of activating BRAF mutations in about 50% of melanoma patients [3,4], the use of selective inhibitors for BRAF and MEK, as well as the development of immune checkpoint inhibitors such as anti-CTLA4 and anti-PD1. Altogether, these developments have resulted in 6-year-survival rates that are presently around 50% [6,30,31,32]. The use of BRAF inhibitors demonstrated the principle susceptibility of melanoma to antiproliferative and proapoptotic therapies, suggesting researchers should search for strategies to further sensitize melanoma cells for apoptosis induction. 

The induction of apoptosis represents a common and efficient way to eliminate tumor cells, and apoptosis deficiency thus plays major roles in drug resistance [13,15]. A number of antiapoptotic factors contribute to apoptosis deficiency in cancer cells, with a special focus on antiapoptotic Bcl-2 proteins, such as Bcl-2, Bcl-x_L_, Bcl-w, and Mcl-1 [33]. Targeting these antiapoptotic Bcl-2 proteins can be achieved by BH3 mimetics, small molecule inhibitors with structural homology to the Bcl homology domain 3 (BH3). Their binding to the hydrophobic groove of antiapoptotic Bcl-2 proteins prevents the ability of antiapoptotic proteins to bind and inhibit proapoptotic family members, thus triggering the induction of apoptosis [34]. Inhibition of anti-apoptotic Bcl-2 family members by BH3 mimetics is a promising strategy for cancer therapy, and BH3 mimetics with specificity for Bcl-2, Bcl-x_L_, and Bcl-w, such as ABT-263, have been tested in clinical trials for patients with hematological malignancies. Meanwhile, the BH3 mimetic ABT-199 (venetoclax), a specific Bcl-2 inhibitor, has been approved by the Food and Drug Administration (2018) and the European Medicines Agency (2021), for clinical use in CLL and AML [33]. 

As for Mcl-1, its chromosomal amplification, as well as increased mRNA and protein levels, were correlated with therapy resistance and sustained growth of different tumors [25,26,35], encouraging the development of specific Mcl-1 inhibitors. Four BH3-mimetics to this protein, S64315, AMG-176, AMG-397, and AZD-5991, have entered clinical evaluation in patients with hematological malignancies, either alone or in combination with venetoclax [36]. The BH3 mimetic and selective Mcl-1 inhibitor S63845, has shown convincing effects on cell viability in a large number of hematological cancer cell lines [27], whereas in melanoma cell lines, high sensitivity (IC50 < 1 µM) and significant effects on cell death, were reported in only a minority (14%) of cell lines [37]. Similarly, the four melanoma cell lines investigated in the present study, revealed only moderate sensitivity to single treatment with 1 µM S63845, as determined by loss of cell viability (max. -35%) and induction of apoptosis (up to 24%). This indicates that Mcl-1 inhibition alone cannot be sufficient for treatment of melanoma. Thus, combination treatments may be helpful.

On the other side, the large therapeutic gaps left by BRAF inhibitors, namely due to BRAF-WT tumors and acquired resistance, demand the development of additional strategies. Targeting of MEK, the downstream MAPK, was considered, however, did not appear to be sufficiently effective when applied alone. Thus, MEK and BRAF inhibitors are presently applied in combination, thus improving response duration and side effects [38,39,40].

BRAF inhibitor resistance is frequently associated with recovered ERK signaling, the main downstream effector in the MAPK pathway [6,22]. Direct ERK inhibition may thus provide a suitable strategy, and several ERK inhibitors have been developed. Thus, LY3214996, BVD-523 (Ulixertinib), MK-8353, and GDC-0994 have been shown to inhibit the growth of melanoma and other solid cancer cells in vitro and in xenograft models [41,42,43,44,45]. Several drugs have already entered clinical trials (phase 1). Thus, partial response and stable disease was reported for Ulixertinib and MK-8353 in 20–33% of patients, however, these were also associated with adverse events such as fatigue, diarrhea, pruritus, and rash [43,44]. 

The selective ERK1/2 inhibitor SCH772984, inhibits the intrinsic kinase activity of ERK, as well as its phosphorylation through MEK [24]. Its activity was demonstrated in BRAF-mutant, NRAS-mutant, and WT melanoma cell lines, as well as in xenograft melanoma models [22,24,46]. As shown here, SCH772984 treatment resulted in early (4 h), as well as sustained, suppression of ERK phosphorylation (24 h), indicating its persistent activity and possibly its ability to overcome resistance. We found, in response to SCH772984, reduced cell viability (down to 60%) and cell proliferation (down to 17%). Importantly, comparable effects were obtained in BRAF-mutated and BRAF-WT melanoma cells, indicating that this therapeutic gap can be overcome. In contrast to cell viability, the effects on apoptosis induction by SCH772984 alone remained on a low level (< 11%), suggesting that the full potential of this inhibitor is not yet utilized in single treatments and that combination strategies could be of advantage.

Drug combinations are frequently applied in approved therapies and in clinical trials to overcome therapy resistance and tumor relapse. Thus, combinations of BRAF and MEK inhibitors have developed as a standard therapy for BRAF-mutated melanoma, and resulted in improved long-term clinical outcomes [38,39,40]. In addition, combinations of anti-PD1 and anti-CTLA-4 antibodies with BRAF and MEK inhibitors have been evaluated, but did not show significant improvement so far, as compared to the currently used combination of BRAF and MEK inhibitors [47,48,49,50]. Other combination strategies have been positively tested in experiments. For instance, melanoma cells show only limited response to the death ligand TRAIL (TNF-related apoptosis-inducing ligand), which can be strongly improved by different treatments used in combination [15]. 

Due to the important roles of antiapoptotic Bcl-2 proteins for tumor cell survival, combinations of inhibitors for antiapoptotic Bcl-2 proteins (BH3 mimetics) with MAPK inhibitors appear to be promising strategies. Thus, BH3 mimetics for targeting Bcl-2, Bcl-x_L_, and Bcl-w (ABT-737, -263, -199) have been developed, and their combination with vemurafenib has resulted in enhanced apoptosis and reduced cell viability in BRAF-mutant melanoma cells [51,52]. As for SCH772984, enhanced effects were reported in response to triple combinations, as in breast cancer cells treated with SCH772984, ABT-737, and HER-2 inhibitors, or in acute myeloid leukemia cells treated with SCH772984, ABT-199, and PI3K/mTOR inhibitors [53,54]. 

The combination of S63845 and vemurafenib has been previously shown to further decrease cell viability in BRAF-mutated melanoma cell lines [27,55]. This is confirmed here for two BRAF-mutated melanoma cell lines at the levels of cell viability, cell proliferation, and apoptosis induction. Interestingly, also two BRAF-WT cell lines showed some response to the combination of vemurafenib/S63845. Previously, less pronounced combination effects on melanoma cell killing were reported, when vemurafenib was used in lower concentrations (3 µM), thus suggesting that relatively high doses of vemurafenib were needed for the positive combination effects [55].

Combinations of S63845 and ERK inhibitors in melanoma have not been reported so far. We found only a report in rhabdomyosarcoma cells, which showed that the combination of S63845 and the ERK inhibitor Ulixertinib, resulted in enhanced apoptosis related to activation of caspases and intrinsic mitochondrial pathways [56]. Here, we demonstrate that the combination of S63845 and the ERK inhibitor SCH772984 resulted in an impressive enhancement of the antitumor effects, both in BRAF-mutated and BRAF-WT melanoma cells. Thus, apoptosis was strongly induced (44–64%) and loss of cell viability was reduced to 7–17%. The effects of SCH772984/S63845 were comparable to those of vemurafenib/S63845 in BRAF-mutated melanoma cell lines.

These strong effects recommend S63845 for combination treatment with MAPK inhibitors, which may apply to approved BRAF inhibitors in BRAF-mutated melanomas, as well as to combinations of SCH772984/S63845 in both BRAF-mutated and BRAF-WT melanomas. Therapeutic efficacy may be significantly enhanced and resistance may be overcome. Although the drug combinations, suggested here, have not been tested so far in vivo, the single treatments are out of the question. Besides vemurafenib and other BRAF inhibitors, which have been in clinical use for many years, the selective ERK1/2 inhibitor SCH772984, has been tested in vivo in xenograft melanoma models [22], and other ERK inhibitors, such as Ulixertinib and MK-8353, have already been tested in phase 1 clinical trials [43,44]. In addition, several Mcl-1 inhibitors have entered clinical evaluation in patients with hematological malignancies [36], of which S64315 (MIK665) is chemically related to S63845 (used here), and has shown comparable activity [57]. Thus, there seems to be good hope that the combination of these drugs may also be tolerated in patients, in particular, as the mutual enhancement shown here may allow a reduction in dose compared to the single treatments.

Concerning the mechanisms by which S63845/SCH772984 induced apoptosis in melanoma cells, we found strong indications of an activation of intrinsic proapoptotic pathways, namely loss of mitochondrial membrane potential, cytochrome c release, as well as caspase-9 and caspase-3 activation, in both BRAF-mutated and WT cells. The essential role of caspases was proven by a pan-caspase inhibitor, which almost completely abolished apoptosis and largely restored cell viability. Activation of caspases and mitochondrial pathways has also been reported in rhabdomyosarcoma cells, in response to the combination of S63845 with the ERK inhibitor Ulixertinib [56]. In melanoma cells, enhanced caspase-3 activation and mitochondrial activation have been reported in response to combinations of S63845 or TW-37 (targeting Mcl-1, Bcl-x_L_ and Bcl-2) with BRAF inhibitors (vemurafenib or encorafenib) [58,59]. 

A role of ROS in the induction of apoptosis by vemurafenib, has been shown by us previously [60]. Here, we show that SCH772984 alone resulted in increased ROS levels, which were further enhanced by combination treatment. A key role of ROS has been suggested in melanoma cells for TW-37, in combination with the inhibition of MEK [61]. The function of ROS could, however, not be proven in the present study, as antioxidants were not effective against ROS produced by combination treatments. There are different types of ROS, which are located in different cellular compartments. Thus, the antioxidative capacity of typical antioxidants, such as NAC and tocopherol, sometimes cannot suffice [62]. We do not speculate that ROS played the decisive role for the drug combinations used here, whereas decisive roles of ROS have been identified in melanoma cells when ROS production appeared as an initial step [18,62]. In the present setting, caspases appeared as an upstream step. Although caspase activation was often reported as being downstream of ROS production and loss of mMP, several data have also suggested the opposite direction under certain conditions, e.g., due to caspase-mediated activation of the proapoptotic Bcl-2 protein Bid [16,17,20]. The relationship between caspase activation and ROS is still largely unclear and remains controversial. However, we have previously shown, in melanoma cells, that ROS production may also come downstream of proteases [62].

As for proteins of the Bcl-2 family, critical regulators of intrinsic apoptosis pathways, we report here the regulation of several proteins in response to SCH772984 or the combination SCH772984/S63845. As for Mcl-1, its upregulation in response to S63845 has already been reported previously, and was also seen here. E.g., Mcl-1 upregulation in response to S63845 was seen in tumor cell lines of breast and colon cancer, rhabdomyosarcoma, and T-ALL [27,56,63,64], and may be understood as a kind of cellular counter-regulation, based on induced Mcl-1 stability [65].

Of note, Mcl-1 upregulation in response to S63845, seen here in two melanoma lines, was completely abrogated in combinations with SCH772984, indicating that Mcl-1 upregulation may be ERK-dependent and that this counter-regulation may be overcome by the combination. Furthermore, we saw some downregulation of Bcl-2 in A-375 by the combination treatment. Downregulation of Mcl-1 or Bcl-2 in response to vemurafenib has also previously been reported in melanoma cells [52,60,66].

The proapoptotic subgroup of BH3-only proteins play particular roles in apoptosis regulation, and their expression is often tightly controlled by MAPK pathways. In particular, Bim and Puma appear to be highly potent BH3-only proteins, simultaneously targeting several antiapoptotic Bcl-2 family members [20,67]. Thus, we found significant upregulation of Bim and Puma in response to SCH772984. However, their increase was not sustained in the combination of S63845/SCH772984, which may be explained by their cleavage through activated caspases in a negative feedback loop [68,69]. Upregulation of Bim and Puma was also reported in melanoma cells in response to BRAF inhibition, which is explained by the suppression of these proteins through active MAPK pathways [20,52,55,60,70].

With respect to the BH3-only protein Noxa, the situations appears to be somewhat more complex. Thus, Noxa was reported as downregulated in response to vemurafenib. Noxa is a direct antagonist of Mcl-1; on the other hand, Bcl-2, Bcl-x_L_, and Bcl-w can be neutralized by Bim and Puma, which are upregulated by MAPK inhibition. Thus, it was suggested that melanoma cells treated with MAPK inhibitors may become particularly dependent on Mcl-1 [58,71,72]. Here, we did not see a downregulation of Noxa by SCH772984, however, it was clearly upregulated in response to combination treatment with S63845/SCH772984. Thus, also the possible lack of Noxa in the course of MAPK inhibition seems to be overcome by this combination.

## 4. Materials and Methods

### 4.1. Cell Culture and Treatment

Two BRAF-mutated cell lines, A-375 [73] and Mel-HO [74], as well as two BRAF-WT melanoma cell lines, MEWO [75] and SK-MEL-23 [76], were cultured at 37 °C, 5% CO_2_ in DMEM (4.5 g/L glucose; Invitrogen, Karlsruhe, Germany), supplemented with 10% fetal calf serum and antibiotics. For control, three different cultures of normal human fibroblasts were used (ATCC, Manassas, VA, USA), which were cultured in RPMI 1640 growth medium + 10% FCS.

Cells were treated with the selective BRAF (V600E) inhibitor vemurafenib/PLX4032 (Selleck Chemicals, Houston, TX, USA), the ERK1/2 inhibitor SCH772984 (Cay19166; Cayman Chemical, Hamburg, Germany), and the Mcl-1 inhibitor S63845 (HY-100741; MedChemExpress, Cologne, Germany), while control cells received the solvent DMSO. For caspase inhibition, the pan-caspase inhibitor QVD-Oph (Abcam, Cambridge, UK; 5 μM) was applied at 1 h before cells were treated with other agents. Most analyses were performed in 24-well plates, and 5 × 10^4^ cells were seeded per well. 

### 4.2. Quantification of Cell Viability and Apoptosis

Cell viability was determined by staining cells with calcein-AM (BD Biosciences, Heidelberg, Germany), which is converted, in viable cells, by intracellular esterases, to green-fluorescent calcein. Cells, grown and treated in 24-well plates, were harvested by trypsinization and stained with 0.5 µM calcein-AM, at 37 °C for 1 h. Thereafter, cells were washed with PBS and evaluated by flow cytometry (FL2H), using a FACS Calibur (BD Bioscience, Bedford, MA, USA).

Quantification of apoptosis was performed by cell cycle analysis. Cells were harvested by trypsinization and lysed in a hypotonic buffer. Isolated nuclei were stained for 1 h with 40 µg/mL propidium iodide (Sigma-Aldrich Chemie, Taufkirchen, Germany). Cells in G1 (gap 1), G2 (gap 2), and S-phase (synthesis), and sub-G1 cells were quantified by flow cytometry (FL3A). The sub-G1 cell population corresponds to cells with fragmented DNA = apoptotic cells. This is due to the washing out of small DNA fragments from apoptotic nuclei, resulting in nuclei with less DNA than in G1 (sub-G1).

Cell death was further quantified by staining cells with Annexin V-fluorescein isothiocyanate (AnnV-FITC) and propidium iodide (PI). Cells harvested by trypsin/EDTA were washed with cold PBS and resuspended at 10^6^ cells/mL in 10 mM Hepes (pH 7.4), 140 mM NaCl, and 2.5 mM CaCl_2_. For 100 µL of cell suspension (10^5^ cells), 5 µL of AnnV-FITC (BD Biosciences, Heidelberg, Germany), and 10 µL of PI stock solution (50 μg/mL, Sigma-Aldrich) were added, and cells were incubated for 15 min at room temperature. Subsequently, samples were analyzed using a FACSCalibur and the CELLQuest software (Becton Dickinson, Heidelberg, Germany).

### 4.3. Cell Proliferation Assays

Relative cell proliferation rates were determined by WST-1 assay (Roche Diagnostics, Penzberg, Germany), based on the staining of cells with the water-soluble tetrazolium salt WST-1. In metabolically active cells, WST-1 is converted to formazan dye by mitochondrial dehydrogenases. The assay was quantified in an ELISA reader at 450 nm. As the enzyme activity is restricted to viable cells, the read-out reflects both cell numbers and cell viability. Thus, reduced WST-1 values may reflect either fewer cells (reduced cell proliferation) or less mitochondrial enzyme activity in single cells (less viable).

### 4.4. Determination of Mitochondrial Membrane Potential and Reactive Oxygen Species (ROS)

Mitochondrial membrane potential (mMP) was determined by staining cells with the fluorescent dye TMRM^+^ (Sigma-Aldrich Chemie). Cells, grown and treated in 24-well plates, were harvested by trypsinization and stained for 20 min, at 37 °C, with TMRM^+^ (1 µM). After being washed two times with PBS, cells were evaluated by flow cytometry (FL2H). 

For microscopic visualization of loss of mMP, and of morphological changes in the course of apoptosis, cells were seeded in 6-well plates (2 × 10^5^ cells/2 mL) and treated for 24 h. Thereafter, cells were incubated for 30 min with 2 µg/mL JC-1 (5,5′,6,6′-tetrachloro-1,1′,3,3′-tetraethyl-benzimidazolylcarbocyanin iodide; Life Technologies, Carlsbad, CA, USA) and with 0.2 µg/mL Hoechst-33342 (Sigma-Aldrich Chemie). After staining, micrographs were taken at different times, using an Axiovert 200 inverse fluorescence microscope (Carl Zeiss, Jena, Germany), equipped with appropriate fluorescence filter sets, and a Hamamatsu ORCA-ER digital camera.

For determination of intracellular ROS levels, cells grown in 24-well plates were pre-incubated for 1 h with the fluorescent dye H_2_DCF-DA (D-399, Thermo Fisher Scientific, Hennigsdorf, Germany, 10 µM), before starting treatment with agonists. At 24 h of treatment, cells were harvested by trypsinization, washed with PBS and analyzed by flow cytometry (FL1H). As positive controls, H_2_O_2_ (1 mM, 1 h), as well as an indirubin derivative (DKP-073, 10 µM, 24 h, [18]), were applied. For ROS scavenging, cells were pretreated for 1 h with 1 mM N-acetylcysteine (NAC, Sigma-Aldrich, Taufkirchen, Germany) or with 1 mM α-tocopherol (vitamin E, Fluka, Steinheim, Germany). 

### 4.5. Caspase Activation Assay and Cytochrome C Release

Cells with activated caspases were detected by using a FLICA Apoptosis Detection kit (Immunochemistry Technology, LLC, Bloomington, MN, USA), following the manufacturer’s instructions. The kit employs carboxyfluorescein-labeled fluoromethyl ketone peptide inhibitors of caspase-8 (FAM-LETD-FMK), caspase-9 (FAM-LEHD-FMK), and caspase-3/7 (FAM-DEVD-FMK), respectively, which are cell-permeable and non-cytotoxic fluorochroms. The inhibitors covalently bind to active caspases, producing green fluorescence, which allows the determination of the percentage of cells with activated caspases. In brief, harvested cells were incubated in FLICA working solution for 1 h (37 °C, 5% CO_2_). Thereafter, they were centrifuged, re-suspended in PBS, and analyzed using a FACSCalibur and CELLQuest software (Becton Dickinson, Heidelberg, Germany).

For determination of cytochrome c release, cytosolic and mitochondrial extracts were separated. Cells harvested in PBS were incubated for 3 min on ice in a hypotonic buffer containing 20 mM HEPES, pH 7.4, 10 mM KCl, 2 mM MgCl_2_, 1 mM EDTA, 0.1 mM PMSF, and 0.025 mg/mL Digitonin (Sigma-Aldrich). After centrifugation, the supernatant was collected as the cytosolic fraction. The pellet was again resuspended in (10 mM Tris-HCl, pH 7.5, 137 mM NaCl, 1% Triton X-100, 2 mM EDTA, 1 µM pepstatin, 1 µM leupeptin, 0.1 mM PMSF) and was kept on ice for 30 min. After another centrifugation, the supernatant was collected as the mitochondrial fraction. Both series were analyzed by Western blotting using a cytochrome c antibody (BD Biosciences, Heidelberg, Germany, 556433, mouse, 1:1000).

### 4.6. Western Blotting

For Western blotting, total protein extracts were obtained in cell lysis buffer containing 150 mM NaCl, 1 mM EDTA, 1% NP-40, 50 mM Tris-HCl (pH 8.0), as well as phosphatase and protease inhibitors. Following SDS polyacrylamide gel electrophoresis, proteins were blotted on nitrocellulose membranes (A15033264, GE Healthcare Life science, Chicago, IL, USA). 

Primary antibodies of Cell Signaling (Danvers, MA, USA): cleaved caspase-3 (9661, rabbit, 1:1000), caspase-8 (9746, mouse, 1:1000), caspase-9 (9502, rabbit, 1:1000), Mcl-1 (4572, rabbit, 1:1000), total-ERK (4695, rabbit, 1:1000), phospho-ERK (9106, mouse, 1:1000), total-Bad (9292, rabbit, 1:1000), phospho-Bad (9296, mouse, 1:1000), Bcl-2 (15071, mouse, 1:1000), Bax (2772, rabbit, 1:1000), Bim (C34C5, rabbit, 1:1000), PARP (9542, rabbit, 1:1000). Primary antibodies of Santa Cruz Biotech (Dallas, TX, USA): Puma (sc-374223, mouse, 1:200), Noxa (N-15, sc-26917, goat polyclonal, 1:200), GAPDH (sc-32233, mouse, 1:200). Other primary antibodies: Total caspase-3 (R&D Systems, Wiesbaden, Germany, AF605, goat, 1:2000), cytochrome c (BD Biosciences, Heidelberg, Germany, 556433, mouse, 1:1000), γH2AX (Ser139) (Millipore, Darmstadt, Germany, 05-636, mouse, 1:1000). β-actin (Sigma-Aldrich, A2066, rabbit, 1:1000). Secondary antibodies: peroxidase-labelled goat anti-rabbit and goat anti-mouse (Dako, Hamburg, Germany; 1:5000), as well as rabbit anti-goat (SouthernBiotech, Birmingham, AL, USA, 1:5000).

### 4.7. Statistical Analyses

The results of all quantitative assays were generally proven by at least three independent experiments. Each individual experiment consisted of triplicate values (three wells that were seeded, treated, and analyzed individually). Thus, for statistical analysis, there were at least nine values in each group. Using these values, mean values and standard deviations (SDs) were determined, and are shown in the figures. For the calculation of statistical significance, Student’s *t*-test was applied, and statistical significance is indicated by asterisks in the figures (* *p* < 0.05). Western blot experiments were repeated at least once, by using two independent series of protein extracts. Signals of Bcl-2 proteins, pERK, PARP, γH2AX and cytochrome c were quantified by densitometric analysis, and were normalized by the respective GAPDH or β-actin values. Thus, median induction factors or the percent of downregulation were calculated from two independent experiments.

For the identification of synergistic effects in combinations of SCH772984 and vemurafenib with S63845, the data of three concentrations of each drug were used, and applied to the web application SyngeryFinder 3.0 [28]. In particular, the ZIP (zero interaction potency) scoring method was used. Accordingly, delta scores of (δ ≥ 10) were considered as synergistic, whereas scores between -10 and 10 were considered as additive.

## 5. Conclusions

Still large gaps remain to be closed in the treatment of metastasized melanoma patients. We thus investigated here the ERK inhibitor SCH772984, as an alternative strategy for BRAF inhibition, as well as combinations with the Mcl-1 inhibitor S63845. The study underlined that BRAF inhibitor resistance, due to lacking a BRAF mutation, may be overcome by ERK inhibitors. However, both the established BRAF inhibitor vemurafenib and the ERK inhibitor SCH772984 showed only rather limited efficacy, when applied alone, as determined by quantification of loss of cell viability and induced apoptosis. 

This dramatically changed when MAPK inhibitors were combined with S63845, the effects of vemurafenib were strongly enhanced in BRAF-mutated cells, and the effects of SCH772984 were strongly enhanced in both BRAF-mutated and BRAF-WT cells. Downstream signaling cascades were related to activation of caspase cascades and proapoptotic mitochondrial pathways, as well as upregulation of several well-known proapoptotic BH3-only proteins of the Bcl-2 family, such as Bim, Puma, and Noxa. Thus, the combination with S63845 may be considered for already established BRAF inhibitors in BRAF-mutated melanoma, and the combination of SCH772984 with S63845 could be a new therapeutic option for all melanomas.

## Figures and Tables

**Figure 1 ijms-24-04961-f001:**
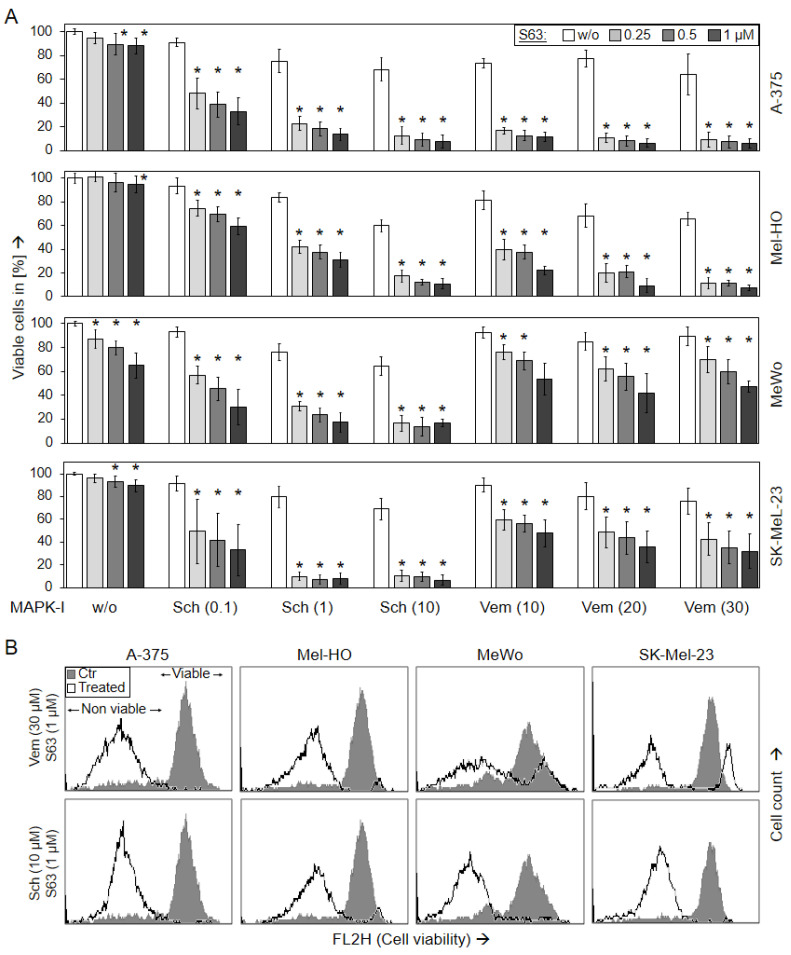
Loss of cell viability (48 h). Melanoma cell lines A-375, Mel-HO, MeWo, and SK-Mel-23 were seeded in 24-well plates, and were treated with S63845 (S63, 0.25, 0.5, 1.0 µM), SCH772984 (Sch, 0.1, 1.0, 10 µM), and vemurafenib (Vem, 10, 20, 30 µM), as indicated. (**A**) After 48 h, cell viability was determined by calcein-AM staining and flow cytometry. Values represent the percentage of cells with high calcein staining (=viable cells). Effects on cell viability were calculated as percentage of non-treated controls (100%). At least three independent experiments were performed, each one consisting of triplicate values. Mean values of all individual values (>8) are shown here. Statistical significance is indicated by asterisks (*p* < 0.05), and was calculated for S63845 treatment as compared to non-treated control cells. Statistical significance of the effects in combination-treated cells was calculated as compared to the respective single treatments (SCH772984 alone or vemurafenib alone, white bars). (**B**) Examples of flow cytometry measurements of cells treated for 48 h with (30 µM vemurafenib/1 µM S63845) or (10 µM SCH772984/1 µM S63845) are shown as overlays vs. controls (Ctr). Non-viable and viable cell populations are indicated.

**Figure 2 ijms-24-04961-f002:**
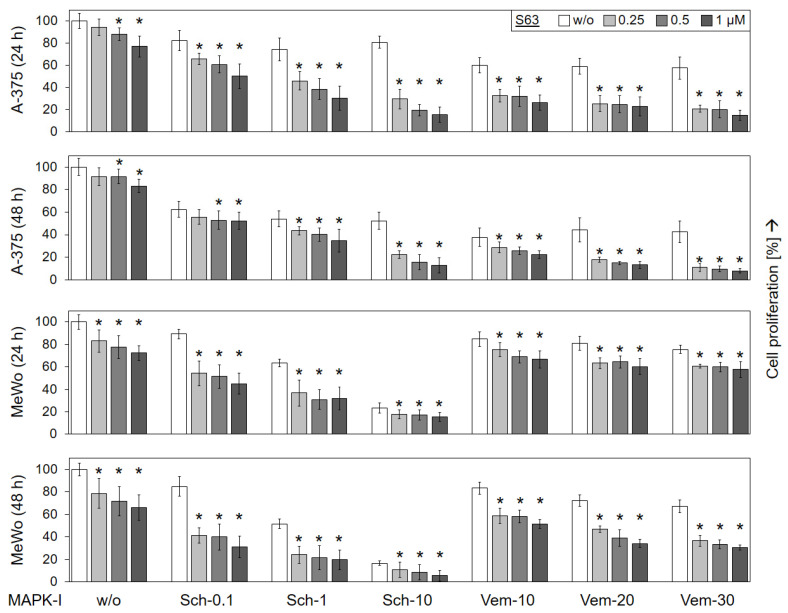
Decreased cell proliferation. Cell lines A-375 and MeWo were seeded in 96-well plates and were treated with increasing concentrations of S63845 (S63, 0.25, 0.5, 1.0 µM), SCH772984 (Sch, 0.1, 1.0, 10 µM) and vemurafenib (Vem, 10, 20, 30 µM), as indicated. Cell proliferation was quantified at 24 h and at 48 h of treatment by WST-1 assay. Effects on cell proliferation were calculated as percentage of non-treated controls (100%). Three independent experiments were performed, each one consisting of triplicate values. Mean values of all individual values (9) are shown here. Statistical significance is indicated by asterisks (*p* < 0.05), and was calculated for S63845 treatment as compared to non-treated control cells. Statistical significance of the effects of combination-treated cells was calculated as compared to the respective single treatments (SCH772984 alone or vemurafenib alone, white bars).

**Figure 3 ijms-24-04961-f003:**
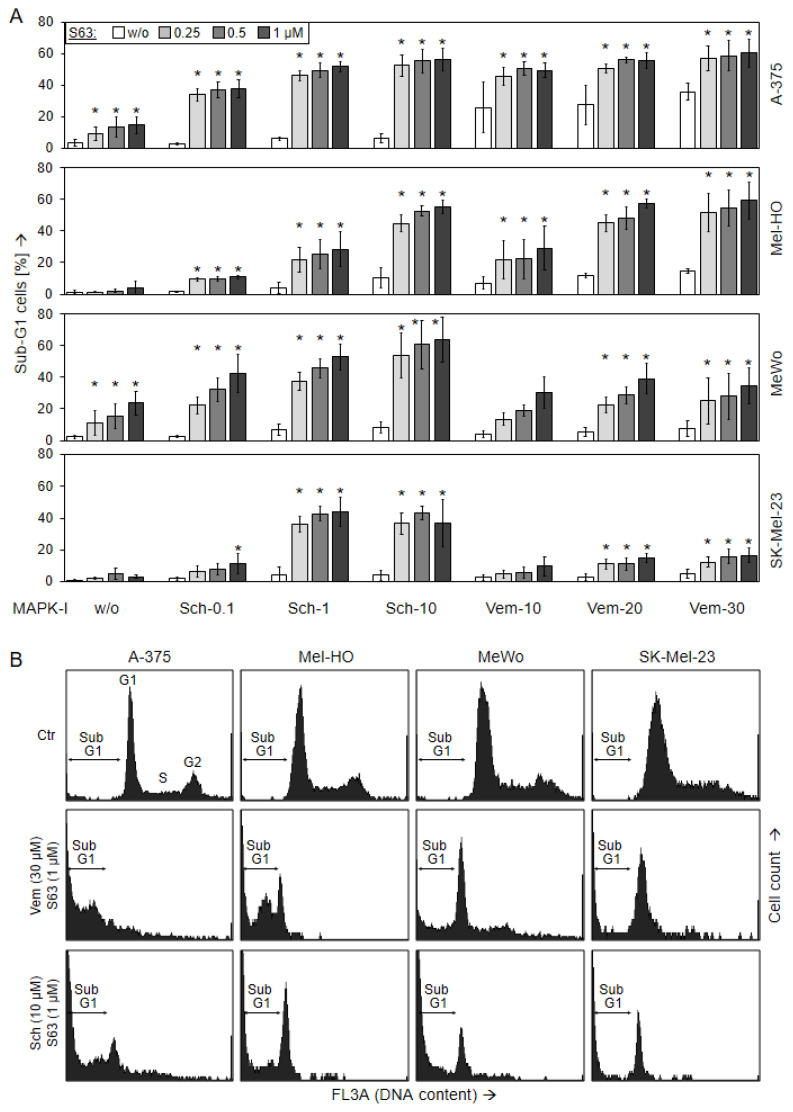
Induction of apoptosis, as shown by cell cycle analysis. Melanoma cell lines A-375, Mel-HO, MeWo, and SK-Mel-23 were seeded in 24-well plates and were treated with S63845 (S63, 0.25, 0.5, 1.0 µM), SCH772984 (Sch, 0.1, 1.0, 10 µM), and vemurafenib (Vem, 10, 20, 30 µM), as indicated. (**A**) After 48 h, apoptotic cells were identified by flow cytometry after propidium iodide staining and flow cytometry as sub-G1 cells (cell cycle analyses). At least three independent experiments were performed, each one consisting of triplicate values. Mean values of all individual values (>8) are shown here. Statistical significance is indicated by asterisks (*p* < 0.05), and was calculated for S63845 treatment as compared to non-treated control cells. Statistical significance of the effects of combination-treated cells was calculated as compared to the respective single treatments (SCH772984 alone or vemurafenib alone, white bars). (**B**) Examples of flow cytometry measurements of cells treated for 48 h with (30 µM vemurafenib/1 µM S63845) or (10 µM SCH772984/1 µM S63845) are shown as compared to controls (Ctr). Cell cycle phases G1 (gap 1), S (synthesis), G2 (gap 2), as well as sub-G1 cells are indicated.

**Figure 4 ijms-24-04961-f004:**
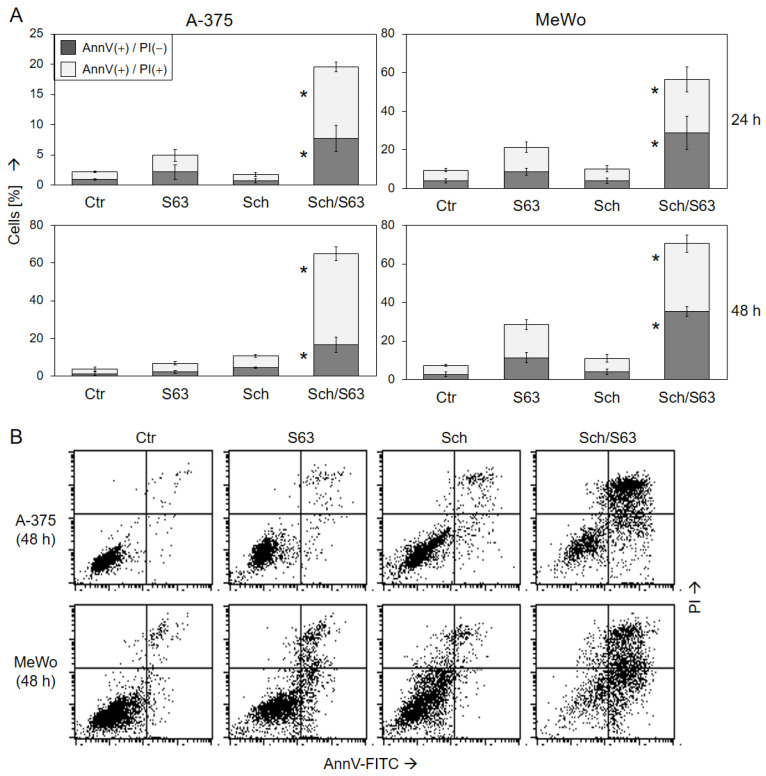
Induction of apoptosis as shown by Annexin V/PI staining. A-375 and MeWo cells were treated with 1 µM SCH772984 (Sch), 1 µM S63845 (S63), or the combination. Cell death analyses by AnnV/PI staining and flow cytometry were performed after 24 h and 48 h. (**A**) Mean values and SDs of two cell death fractions, namely AnnV(+)/PI(−) and AnnV(+)/PI(+) cells, were calculated from four independent experiments, each one consisting of triplicate values, and are shown as %. Statistical significance of the effects of combination treatments, as compared to controls and as compared to single treatments, was calculated by *t*-test, and is indicated by asterisks for both cell death fractions (*p* < 0.01). (**B**) Representative flow cytometry histograms of treated and control cells, separated in four quadrants (according to PI and AnnV positivity), are shown below.

**Figure 5 ijms-24-04961-f005:**
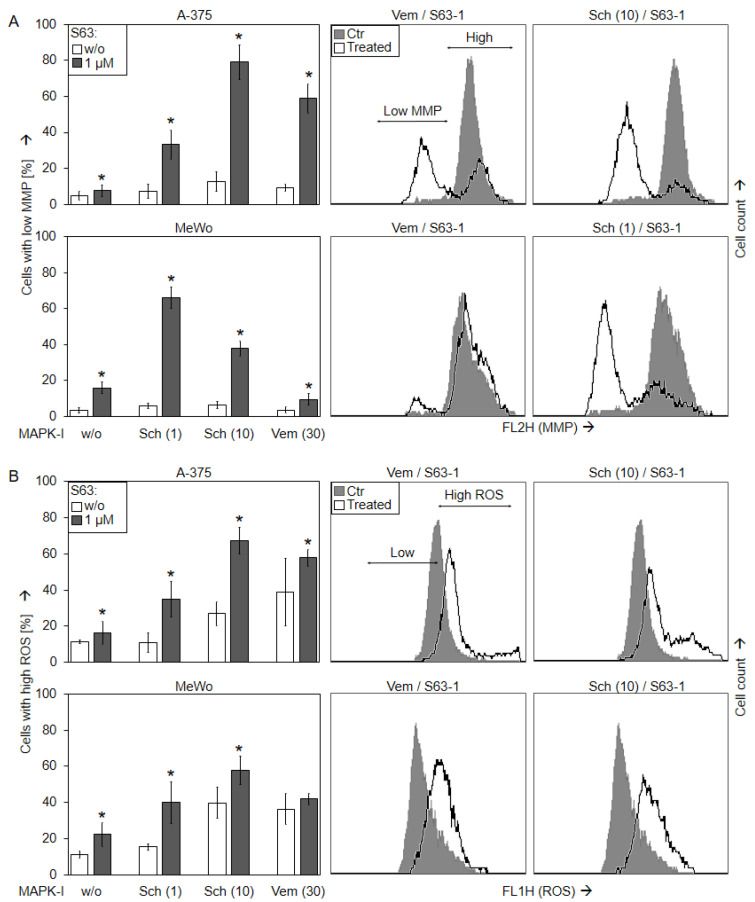
Loss of mMP and ROS production. A-375 and MeWo cells were treated with S63845 (S63, 1 µM), SCH772984 (Sch, 1, 10 µM) and/or vemurafenib (Vem, 30 µM). (**A**) Mitochondrial membrane potential (mMP) was determined at 24 h by TMRM^+^ staining and flow cytometry. Values represent the percentage of cells with loss of mMP. Examples of flow cytometry readings are shown on the right side for combination treatments of vemurafenib/S63845 and SCH772984/S63845 (overlays vs. controls). Cell populations with low and high (normal) mMP are indicated. (**B**) Cellular levels of ROS were determined by H_2_DCF-DA staining and flow cytometry. Values represent the percentage of cells with high ROS. Examples of flow cytometry readings are shown on the right side for combination treatments of S63845/vemurafenib and S63845/SCH772984 (overlays vs. controls). Cell populations with low (normal) and high ROS are indicated. (**A**,**B**) Mean values and SDs were calculated from three to four independent experiments, each one consisting of triplicate values, and are shown in %. Statistical significance, indicated by asterisks (*p* < 0.05), was calculated for S63845 as compared to non-treated control cells, whereas the effects of combination-treated cells were compared to the respective single treatments (SCH772984 or vemurafenib alone, white bars).

**Figure 6 ijms-24-04961-f006:**
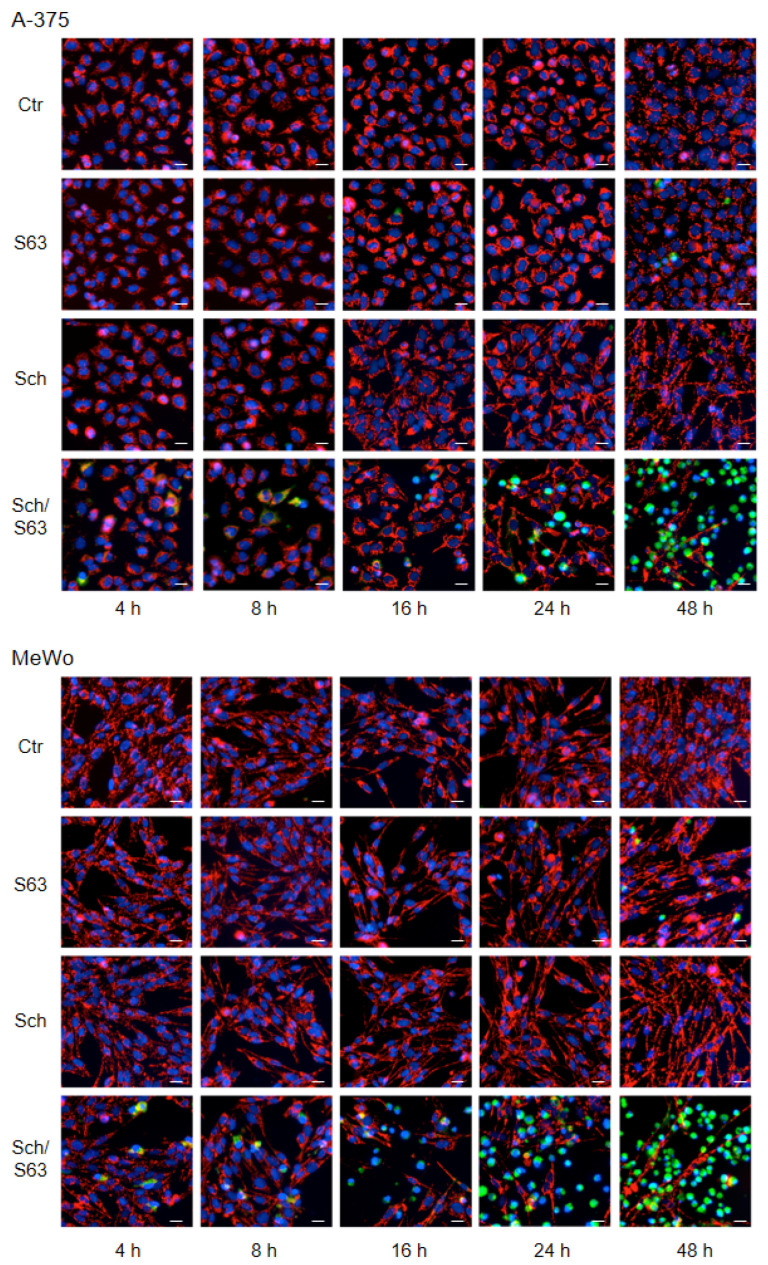
Loss of mMP, as shown by JC-1 staining. A-375 and MeWo cells were treated with SCH772984 (Sch, 1 µM), S63845 (S63, 1 µM), or the combination. For microscopic visualization of loss of mMP, cells were stained with JC-1 and counterstained with Hoechst-33342 at 4, 8, 16, 24, and 48 h of treatment. Blue, nuclear staining; red, mitochondria with high (normal) mMP; faint green, JC-1-stained cytosol; bright green or turquoise, rounded and detached cells with loss of mMP. White scale bars are shown below, right (20 µm). One of two independent experiments, which revealed highly comparable results, is shown.

**Figure 7 ijms-24-04961-f007:**
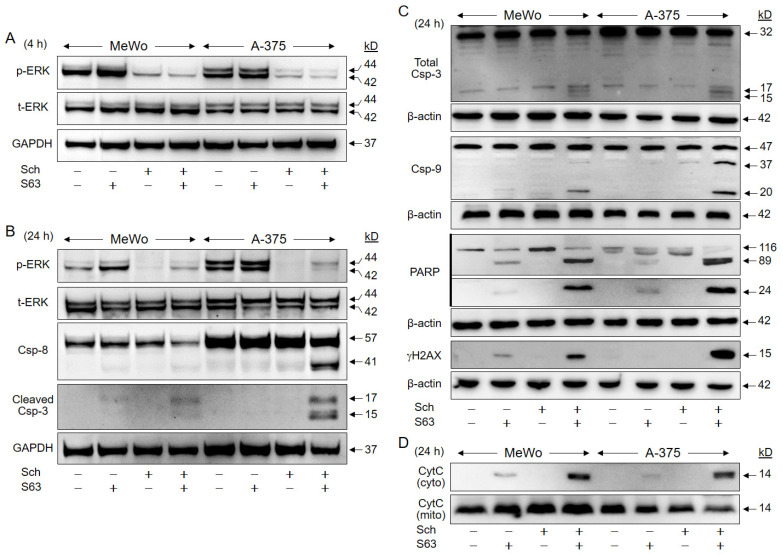
Activation of proapoptotic pathways. A-375 and MeWo cells were treated with SCH772984 (Sch, 1 µM), S63845 (S63, 1 µM), or the combination. Protein extracts were analyzed by Western blotting for ERK phosphorylation (p-ERK) and total ERK expression (t-ERK) at 4 h (**A**), and at 24 h (**B**). Processing of caspase-3, caspase-8 and caspase-9 (Csp) was analyzed at 24 h. For caspase-3, two different antibodies against cleaved caspase-3 (**B**) and total caspase-3 (**C**) were applied. In addition, cleavage of PARP (89, 24 kDa), and phosphorylation of histone H2AX (γH2AX), were analyzed (**C**). For determination of mitochondrial cytochrome c release, cytosolic (cyto) and mitochondrial (mito) cell fractions were separately analyzed (**D**). Equal protein amounts (30 µg per lane) were separated by SDS-PAGE, and consistent blotting was proven by Ponceau staining, as well as by expression of GAPDH and β-actin, respectively. Molecular weights are indicated in kD. Each two independent series of protein extracts and Western blots revealed highly comparable results.

**Figure 8 ijms-24-04961-f008:**
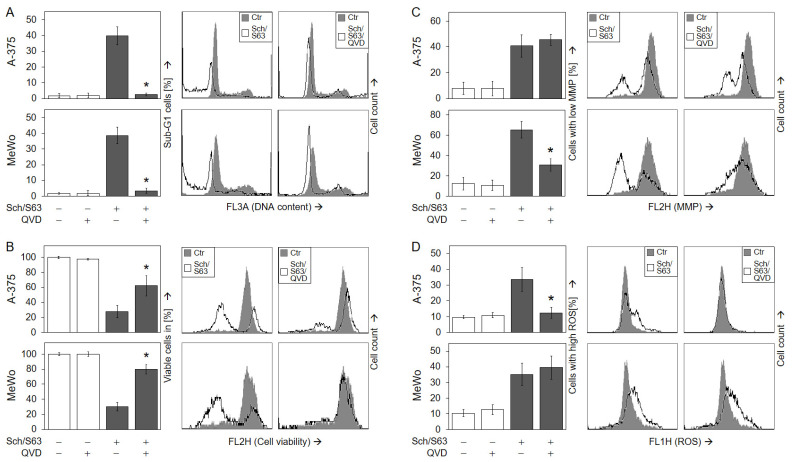
Significant role of caspase activation. A-375 and MeWo cells were treated for 24 h with the combination of SCH772984 (Sch, 1 µM) and S63845 (S63, 1 µM). In addition, cells received the pan-caspase inhibitor QVD-Oph (QVD, 10 µM), when indicated; QVD-Oph was applied at 1 h before other treatments started. The inhibitory effects of QVD on apoptosis induction (**A**) and cell viability (**B**) were determined at 48 h, while the effects on mMP (**C**) and on ROS (**D**) were determined at 24 h. (**A**–**D**) Mean values (in %) and SDs were calculated from three independent experiments, each one consisting of triplicate values. Statistical significance of QVD treatments as compared to SCH772984/S63845 was calculated from all individual values (9), and is indicated by asterisks (*p* < 0.05). On the right side, example flow cytometry readings after SCH772984/S63845 combination treatment, +/− QVD, are shown as overlays with control cells.

**Figure 9 ijms-24-04961-f009:**
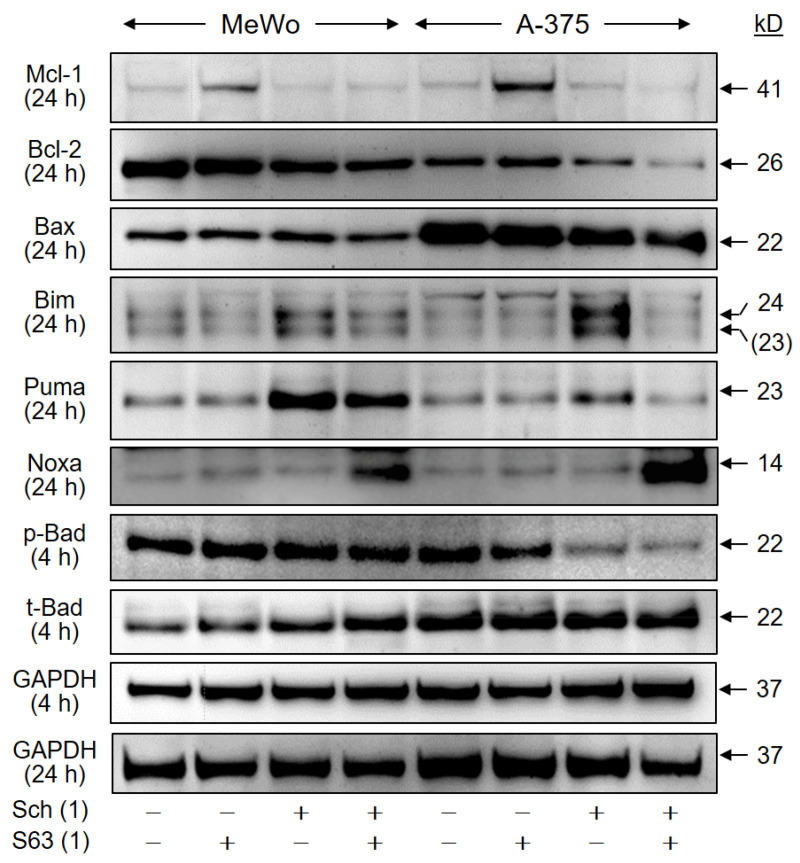
Regulation of Bcl-2 proteins. Cell lines MeWo and A-375 were treated with SCH772984 (Sch, 1 µM), S63845 (S63, 1 µM), or the combination. Protein extracts were analyzed at 24 h by Western blotting, for expression of Mcl-1 (41 kD), Bcl-2 (26 kD), Bax (22 kD), Bim_EL_ (24 kD), Puma (23 kD), and Noxa (14 kD). Expression of phosphorylated and total Bad protein (pBad, t-Bad, 22 kD) was analyzed at 4 h of treatment. Equal protein amounts (30 µg per lane) were separated by SDS-PAGE, and consistent blotting was proven by Ponceau staining, as well as by evaluation of GAPDH expression. Molecular weights are indicated in kD. Two independent series of protein extracts and Western blots revealed highly comparable results.

## Data Availability

All data generated during this study are included in this published article and its Appendix A.

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
