# Peer review of "Enhanced Apoptosis and Loss of Cell Viability in Melanoma Cells by Combined Inhibition of ERK and Mcl-1 Is Related to Loss of Mitochondrial Membrane Potential, Caspase Activation and Upregulation of Proapoptotic Bcl-2 Proteins"

_ijms, 2023, doi:10.3390/ijms24054961_

Round 1

Reviewer 1 Report

Introduction well described. Materials and methods adeguated. Results described. Conclusion adeguated. references adeguated

Author Response

Reviewer 1 - comments and response:

Comments

Introduction well described, Materials and methods adequated, Results described, Conclusion adequated, References adequated.

Response: Thank you very much for the positive evaluation.

Reviewer 2 Report

The authors provide evidence that therapeutic use of a combination of drugs targeting different proteins of signaling cascades including the apopoptic pathway is much more efficient than using only one single substance. Moreover, a combined therapy would reach a far larger number of mutants/different melanoma cells. The data presented are promising, in particular when recalling the big disappointment caused by Verumafenib treatment. Verumafenib, even though a malignant melanoma was identified with the BRAF(VE1000) mutation, did not affect all cells, because a very, very few (but enough to eventually be lethal) did not respond to Verumafenib. Exactly these cells could be addressed by additional components of a drug cocktail. 

The experimental approach is sound, the paper is well written, the experiments are clear and match. Based on the findings, the conclusions drawn are logic and absolutely justified.

There are two points I came across:

1. How would healthy cells, e.g. melanocytes, respond to such a drug cocktail. Would they also increase their apoptotic rate? And if so, would this be an acceptable side effect?

2. Two independent experiments is quite a low number, even if each one was performed as triplicate. What do the error bars represent? The standard deviation (SD) or the standard error (SEM)? If the mean values of the two independent experiments are to be compared, SEM should be given, but then the number of trials would not be sufficient.   

Minor: 

1. Page 3, 2.4, first line: The abbreviation MMP might be confusing, especially when talking about cancer cells, migration and invasion, because MMP is also used as abbreviation for matrix metalloprote(in)ase(s). What about "mMP" instead? 

2. Figure legend 3B, "...are shown as overlays vs. controls". I cannot find/see the overlay (as compared to 1B or 5B where they are clearly visible). 

3. page 14, 3rd last line vs. Figure legend 9, third line: BIM(EL) vs BIM(L). Is there a difference between these two BIMs?

4. page 18, line 5,6: "Intrinsic apoptosis pathways are criticially controlled by the family of pro- and antiapoptotic Bcl-2...". This information could be given much earlier in the Discussion section, because already on page 17 (pro-/anti-)apoptic proteins, apoptosis itself and its therapeutic induction are discussed. 

Typos, errors, style:

- page 2, lines 15 ("..., as tumor necrosis factor-alpha...") and 18 ("effector caspases as caspase 3"), suggestion: use "such as" instead of just "as".

- page 3, 2nd last line: "microscopy images" = "micrographs"?

- page 4, 2.6., line 2: "Each individual experiment was consisted of...", "was" can be removed here.

- page 6, 3.2, 2nd paragraph, line 2, suggestion: insert a comma between "Thus" and "in BRAF-mutated A375, cell proliferation rates were decreased....

- same paragraph, 2nd last sentence, either insert two commas (1. between "...A-375"  and "decreased at 24/48 h....", and 2. between "...by 10 µM SCH772984" and "further dropped to 19/17%..."), or, instead of the two commas, insert "and" between "SCH772984" and "further dropped to 19/17%..."). 

- Page 15, beginnig of discussion, some authors prefer "While" instead of "Whereas" at the beginning of a sentence.

- Similarly, page 17, line 11, and page 18, line 9, abbreviation "E.g." at the beginning of a sentence. Maybe, replace "E.g.,.." with "For example,...." or "For instance,....".

Author Response

Reviewer 2 - Comments and response

General: The authors provide evidence that therapeutic use of a combination of drugs targeting different proteins of signaling cascades including the apopoptic pathway is much more efficient than using only one single substance. Moreover, a combined therapy would reach a far larger number of mutants/different melanoma cells. The data presented are promising, in particular when recalling the big disappointment caused by Verumafenib treatment. Verumafenib, even though a malignant melanoma was identified with the BRAF(VE1000) mutation, did not affect all cells, because a very, very few (but enough to eventually be lethal) did not respond to Verumafenib. Exactly these cells could be addressed by additional components of a drug cocktail.

The experimental approach is sound, the paper is well written, the experiments are clear and match. Based on the findings, the conclusions drawn are logic and absolutely justified.

Response: Thank you very much for this positive evaluation as well as for the subsequent good and helpful suggestions.

Comment 1. How would healthy cells, e.g. melanocytes, respond to such a drug cocktail. Would they also increase their apoptotic rate? And if so, would this be an acceptable side effect?

Response: The question of possible side effects and how normal cells may respond to the drugs is of course important. Following the reviewer´s suggestion, we investigated in additional experiments three independent cultures of normal human fibroblasts. Assays for apoptosis and cell viability were performed at 48 h of treatment. Cultures of normal human melanocytes are presently not available in our lab.

Furthermore, we had tested in a previous study the effects of S63845 in normal peripheral blood mononuclear cells of healthy donors (PBMCs) (Sumarni et al., 2022; Int J Mol Sci 23:12471).

According to the new data, normal fibroblasts responded much less to the combination of S63845 / SCH772984 (each 1 µM) as compared to the melanoma cells, revealing a reduced cell viability of only 82% +/- 7% and an induction of apoptosis of only 11% +/-2%. As there are already nine figures in the manuscript, we suggest to show the fibroblast data in an additional supplementary figure (new Fig S4).

As concerning the effects of S63845 alone in PBMCs, we had previously shown that PBMC cultures were highly sensitive to treatment with 1 µM S63845 alone. However, these cells were also highly sensitive to the Bcl-2/Bcl-w/Bcl-xL antagonists ABT-263 (1 µM) and ABT-737 (1 µM) (Sumarni et al., 2022; Int J Mol Sci 23:12471).

While the fibroblast data support the major idea of the present manuscript, the PBMC data may indicate possible side effects. However, it remains an open question whether the response of cultured PBMCs ex vivo may really reflect a similar response in the in vivo situation. Thus, S63845 was decribed as largely well tolerated in mice with no significant weight loss observed (Kotschy et al., 2016; Nature 538:477-482), and it is currently even in a phase I clinical trial (Wang et al., 2021; J Hematol Oncol 14:67). Also for ABT-263, suitable safety profiles were reported in phase I/II clinical trials for hematological malignancies [de Vos et al., 2021; Leuk Lymphoma, 62, 810-818], despite the strong response seen ex vivo. Cultured PBMCs in growth medium may lack survival signals that are supplied in the blood. Additional animal experiments and finally clinical testing may give a better answer to this question.

Changes:

Culture of normal human fibroblasts is described in materials and methods (2.1.Cell culture and treatment).

Response of fibroblasts to S63845 / SCH772984 is explained in Results section “3.3. Decreased cell viability correlates with induction of apoptosis” (last paragraph) as well as in the supplementary Fig S4.

The question of possible side effects is discussed (Discussion, §11, page 18).

Comment 2. Two independent experiments is quite a low number, even if each one was performed as triplicate. What do the error bars represent? The standard deviation (SD) or the standard error (SEM)? If the mean values of the two independent experiments are to be compared, SEM should be given, but then the number of trials would not be sufficient.  

Response: Our data were generally based on at least two experimental series of experiments, often 3-4 series. Each series itself was consisted of three independent values. I want to indicate that these triplicates are not just triplicate readings or something like that. The triplicates were always completely individual wells, seeded, treated, harvested and analyzed separately. Thus, we considered these as three independent values.

Nevertheless, to put the data on an even more solid statistical basis, we followed the suggestion of the reviewer and performed a large number of additional repeats of the experiments. Thus, now we have for all quantitative assays (Figs 1, 2, 3, 4, 5, 8) at least three independent series of experiments with each three independent values (together 9-12 values). These values were used for new calculations of means and SDs. Repeating so many experiments was a lot of work, thus the revision needed longer time.

The effects of the combination treatments vs. single treatments are generally very strong. Furthermore, all three S63845 concentrations showed comparable combination effects, which confirmed each other. Furthermore, largely comparable combination effects on cell viability, cell proliferation and apoptosis were found in all four cell lines.

Thus, synergistic enhancement of the effects of MAPK inhibitors by the Mcl-1 inhibitor was proven in multiple experiments and is not really in question. Nevertheless, now we have always at least three independent experiments. Synergism of the combinations SCH772984/S63845 and vemurafenib/S63845 was further calculated by the program SyngeryFinder 3.0 [Ianevski et al., 2022]. The combination SCH772984/S63845 turned out as synergistic in all four cell lines (δ scores > 10), while the combination of vemurafenib/S63845 appeared as strongly synergistic only in the BRAF mutated cell lines (Fig S3).

As concerning the supplementary figures S1 (Viability at 24 h) and S2 (Apoptosis at 24 h), we have only two complete independent series of experiments (6 values). In response to the reviewers comment, we thus omitted here the SDs and asterisks for significance. In the figure legends, we indicate that these are median values of two independent experiments. This should not be problematic, as the effects at 24 h strongly resemble those at 48 h, for which we present here a complete statistics (3-4 independent experiments, 9-12 values).

Changes:

Materials and methods: The statistical procedure is explained (2.7. Statistical analyses).

Fig 1 (4 cell lines, viability, 48 h); Fig 2 (A-375, MeWo, WST-1 assay, 24 h, 48 h); Fig 3 (4 cell lines, apoptosis, 48 h); Fig 4 (Annexin V/PI, A-375, MeWo, 24 h, 48 h); Fig 5 (MMP, ROS, A-375, MeWo, 24 h):

For figures 1 - 5, the complete experiments were repeated one more time. Thus, now there are always at least 3-4 independent series (9-12 values). Using these values, means and SDs were recalculated. The new values are always largely similar to the previous ones. Sometimes, the SD values are somewhat increased, whereas statistical significance was not in question. All figures were actualized and exchanged; the figure legends were actualized and the slightly changed values were adjusted in the results text.

Fig 8 (Effects of QVD): These were already three series of independent experiments (9 values); we just corrected the figure legend accordingly.

Minor:

  1. Page 3, 2.4, first line: The abbreviation MMP might be confusing, especially when talking about cancer cells, migration and invasion, because MMP is also used as abbreviation for matrix metalloprote(in)ase(s). What about "mMP" instead?

Response: we followed this suggestion and changed MMP à mMP, throughout the manuscript.

  1. Figure legend 3B, "...are shown as overlays vs. controls". I cannot find/see the overlay (as compared to 1B or 5B where they are clearly visible).

Response: Thank you for indicating this error. It has been corrected “as compared to” (legend to figure 3).

  1. page 14, 3rd last line vs. Figure legend 9, third line: BIM(EL) vs BIM(L). Is there a difference between these two BIMs?

Response: BimEL (24 kDa) and BimL (21 kDa) represent two described splice products of Bim, which are both proapoptotic. In our Western blots, we found induction of BimEL (24 kDa). Weak expression of BimEL in non-treated cultures of melanoma cells was already previously described by us (Plötz et al., 2013; Cancer Lett 335:100-8. PMID: 23402819).

We clarified this issue in results and corrected the legend to figure 9 (BimL à BimEL).

  1. page 18, line 5,6: "Intrinsic apoptosis pathways are criticially controlled by the family of pro- and antiapoptotic Bcl-2...". This information could be given much earlier in the Discussion section, because already on page 17 (pro-/anti-)apoptic proteins, apoptosis itself and its therapeutic induction are discussed.

Response: We agree with this suggestion. The information, including references, has already been given in the introduction (page 2, paragraph 3). So it´s not really needed this late in the discussion. We therefore fused here two sentences: “As for Bcl-2-related proteins, critical regulators of intrinsic apoptosis pathways, we report here …”

Changes: Discussion, §14, now page 19.

Typos, errors, style:

- page 2, lines 15 ("..., as tumor necrosis factor-alpha...") and 18 ("effector caspases as caspase 3"), suggestion: use "such as" instead of just "as".

Response: has been corrected (Introduction, page 2).

- page 3, 2nd last line: "microscopy images" = "micrographs"?

Response: has been corrected (Materials and Methods, 2.4.Determination of mitochondrial membrane potential and reactive oxygen species (ROS).

- page 4, 2.6., line 2: "Each individual experiment was consisted of...", "was" can be removed here.

Response: has been corrected (Materials and Methods, 2.7.Statistical analyses).

- page 6, 3.2, 2nd paragraph, line 2, suggestion: insert a comma between "Thus" and "in BRAF-mutated A375, cell proliferation rates were decreased....

Response: has been corrected (Results, 3.2.Loss of cell proliferation in parallel with decreased cell viability, page 6).

- same paragraph, 2nd last sentence, either insert two commas (1. between "...A-375"  and "decreased at 24/48 h....", and 2. between "...by 10 µM SCH772984" and "further dropped to 19/17%..."), or, instead of the two commas, insert "and" between "SCH772984" and "further dropped to 19/17%...").

Response: We have changed this sentence to make it easier to understand: “Thus, cell proliferation rates that had been decreased to 82/53 % by 10 µM SCH772984 in A-375 at 24/48 h, further dropped to 19/17 % in response to 10 µM SCH772984 / 1 µM S63845.” (Results, 3.2.Loss of cell proliferation in parallel with decreased cell viability)

- Page 15, beginning of discussion, some authors prefer "While" instead of "Whereas" at the beginning of a sentence.

Response: We exchanged “whereas” for “while” in the first sentence of Discussion as well as in a sentence in Results (3.2.Loss of cell proliferation in parallel with decreased cell viability, pages 6)

- Similarly, page 17, line 11, and page 18, line 9, abbreviation "E.g." at the beginning of a sentence. Maybe, replace "E.g.,.." with "For example,...." or "For instance,....".

Response: We also exchanged "E.g." at this place by using “for instance” (Discussion, §7, page 17).

Reviewer 3 Report

The article submitted by Peng et al., titled “Enhanced apoptosis and loss of cell viability in melanoma cells by combined inhibition of ERK and Mcl-1 is related to loss of mitochondrial membrane potential, caspase activation and up-regulation of proapoptotic Bcl-2 proteins” is very well written, and most of the experiments are done nicely.

In this article the author has shown that in combination with the Mcl-1 inhibitor S63845, effects of vemurafenib are strongly enhanced in BRAF-mutated cell lines, and effects of SCH772984 are enhanced in both BRAF-mutated and BRAF-WT cells.

This resulted in up to 90% loss of cell viability and cell proliferation as well as in induction of apoptosis in up to 70% of cells.  They also observed increased caspase activation, loss of mitochondrial membrane potential, increased ROS generation etc.

The author has observed strongly enhanced effects on cell viability, apoptosis and proapoptotic pathways in drug resistance melanoma after the combined effect of vemurafenib as well as SCH772984 with the Mcl-1 inhibitor S63845.

I have few suggestions to further enhance the quality of the manuscript:

The author has observed induction of apoptosis in upto 70 % of the cells. The apoptosis was measured by AnnexinV/PI based FACS assay. To support the apoptotic pathways, they have measured Caspase activation, mitochondrial membrane potential and ROS.

1.     In ROS assay, H2O2 has been used as a positive control. In all the FACS panel of ROS, the author should add a negative control using N-acetylcysteine (NAC). The experiments should run in presence or absence of NAC.

2.     In apoptotic conditions Caspase activation has been measured. In addition, the cleavage of Caspase substrate such as PARP should be measured by western blotting.

3.     The author has observed change in transmembrane potential of mitochondria by JC-1 dye. After that, it would be better to measure the release of Cytochrome C in the cytosol by western blot or by microscopy using Ab specific to Cyt. C.

4.     The hall mark of apoptosis is DNA fragmentation, that can be measured by gel electrophoresis of by TUNEL.

Author Response

Reviewer 3

Comments and Suggestions for Authors

The article submitted by Peng et al., titled “Enhanced apoptosis and loss of cell viability in melanoma cells by combined inhibition of ERK and Mcl-1 is related to loss of mitochondrial membrane potential, caspase activation and up-regulation of proapoptotic Bcl-2 proteins” is very well written, and most of the experiments are done nicely.

In this article the author has shown that in combination with the Mcl-1 inhibitor S63845, effects of vemurafenib are strongly enhanced in BRAF-mutated cell lines, and effects of SCH772984 are enhanced in both BRAF-mutated and BRAF-WT cells. This resulted in up to 90% loss of cell viability and cell proliferation as well as in induction of apoptosis in up to 70% of cells.  They also observed increased caspase activation, loss of mitochondrial membrane potential, increased ROS generation etc. The author has observed strongly enhanced effects on cell viability, apoptosis and proapoptotic pathways in drug resistance melanoma after the combined effect of vemurafenib as well as SCH772984 with the Mcl-1 inhibitor S63845.

I have few suggestions to further enhance the quality of the manuscript:

The author has observed induction of apoptosis in up to 70 % of the cells. The apoptosis was measured by AnnexinV/PI based FACS assay. To support the apoptotic pathways, they have measured Caspase activation, mitochondrial membrane potential and ROS.

Response: Thank you very much for the positive evaluation as well as for the subsequent comments and suggestions.

  1. In ROS assay, H2O2 has been used as a positive control. In all the FACS panel of ROS, the author should add a negative control using N-acetylcysteine (NAC). The experiments should run in presence or absence of NAC.

Response: We have some experience with the use of NAC and other antioxidants from previous projects. Thus, the indirubin derivative DKP-073 resulted in characteristic and early ROS production in melanoma cells. In this setting, NAC (800 µM) completely abrogated ROS production as well as it completely prevented DKP-073-induced apoptosis (Zhivkova et al., 2019; Mol Carcinog 58:258-269, doi: 10.1002/mc.22924).

Thus, we also tried to prevent ROS production in course of SCH772984/S63845 treatment by using NAC (1 mM), as suggested. As an alternative antioxidative strategy, we also tried to suppress ROS by tocopherol (vitamin E, 1 mM). However, ROS production in course of SCH772984/S63845 treatment could not be prevented by antioxidants, neither by NAC nor by tocopherol (new Fig S6A). In consequence, NAC and tocopherol had no significant effect on SCH772984/S63845-induced loss of cell viability (new Fig S5B) and on induced apoptosis (new Fig S6C). To check the activity of NAC, we used DKP-073 (from the previous project) again as control. Proving the full antioxidative activity of NAC in these experiments, DKP-073-induced ROS production was completely prevented by NAC (Fig S6A).

There are different types of ROS, which are further located at different cellular compartments. Thus, the antioxidative capacity of typical antioxidants as NAC and tocopherol sometimes is not sufficient. We found this also in previous projects, e.g. when melanoma cells were treated with the iron-containing cytosine analogue N69 (Franke et al., 2010; Biochem Pharmacol 79:575-86). We do not think that ROS played the decisive role in the present setting, as ROS was induced only later at 24 h after treatment. For other ROS-related strategies, we reported very early ROS production (< 4 h), e.g. for indiribin and N69, cited above.

Changes:

The use of NAC and tocopherol is described in materials and methods.

The findings are explained in results (3.5.Significant role of caspase activation, last paragraph) and are shown in the new Fig S6.

The negative findings on ROS production at earlier time (4 h) is mentioned as data not shown. 2.4.Determination of mitochondrial membrane potential and reactive oxygen species (ROS).

The role of ROS is briefly addressed in the discussion (§13). A new reference is added here (66 Franke et al., 2010).

  1. In apoptotic conditions Caspase activation has been measured. In addition, the cleavage of Caspase substrate such as PARP should be measured by Western blotting.

Response: Following this suggestion, we also investigated PARP cleavage in A-375 and in MeWo in response to S63845, SCH772984 and the combination treatment. Strong induction of PARP cleavage products (89, 24 kDa) was found upon combination treatment, further underlining induced caspase activity. In contrast, no PARP cleavage was seen in the controls and SCH772984-treated cells, whereas PARP cleavage in response to S63845 was much less pronounced (Fig 7C).

Changes:

Abstract: The effects on PARP are briefly mentioned

Introduction: Caspase-mediated PARP processing is briefly mentioned (§3).

Materials and methods: The used PARP antibody is listed (2.6. Western blotting).

Effects on PARP processing are explained in Results (3.5.Significant role of caspase activation).

Results are displayed in Fig 7C

Legend to figure 7 adjusted

  1. The author has observed change in transmembrane potential of mitochondria by JC-1 dye. After that, it would be better to measure the release of Cytochrome C in the cytosol by western blot or by microscopy using Ab specific to Cyt. C.

Response: Besides the JC-1 staining (Fig 6), we had also reported the loss of mitochondrial membrane potential (Fig 5). Both are clear indications of an activation of proapoptotic mitochondrial pathways.

To further support the significance of this pathway, we now also determined mitochondrial release of cytochrome C in A-375 and MeWo in response to S63845, SCH772984 and the combination treatment, as suggested.

Clearly supporting the data in Fig 5 and Fig 6, we found strong cytochrome c release by the combination treatment, whereas cytochrome c was not detected in controls and SCH772984-treated cells, and was much less pronounced in response to S63845 alone.

Changes:

Abstract: The cytochrome c result is briefly mentioned.

Introduction: The role of cytochrome c release is briefly mentioned (§4).

Materials and methods: The protocol of cell fractionation (cytosolic/mitochondrial) is described, and the used cytochrome c antibody is listed (2.5. Caspase activation assay and cytochrome c release).

Results: The cytochrome c results are explained (3.5.Significant role of caspase activation).

Figure 7 was re-organized, and the new data are shown in Fig 7D.

Legend to figure 7 adjusted

  1. The hall mark of apoptosis is DNA fragmentation, that can be measured by gel electrophoresis of by TUNEL.

Response: We agree that DNA fragmentation is the major hallmark of apoptosis. In fact, the apoptosis assay we used in Figure 3 measures just that. The determination of sub-G1 nuclei in a cell cycle analysis is based on fragmented DNA in apoptotic cells, which is washed out in this protocol, thus leading to nuclei, which have less DNA than G1 cells. The advantage of the sub-G1 assay, as compared DNA analysis by gel electrophoresis, is that the number of cells with fragmented DNA can be determined in a quantitative manner (% of sub-G1 cells = apoptotic cells). This assay is distinct from direct propidium iodide staining of cells, which identifies damaged (cytotoxic) cells.

However, to further address the question of the reviewer, we also determined the phosphorylation of histone H2AX at position Ser-139, denoted as γH2AX. This is a well-accepted marker of DNA fragmentation and DNA double strand breaks. Thus, DNA fragmentation induced by apoptosis results in considerable induction of DNA double strand breaks. This in turn induces the phosphorylation of H2AX termed γH2AX (Rogakou et al. 2000; J Biol Chem 275: 9390-9395).

Western blotting revealed strong induction of γH2AX in MeWo and A-375 cells in course of SCH772984/S63845 combination treatment, whereas γH2AX was not detected in control cells and in cells treated by the ERK inhibitor alone. Thus, this assay strongly confirmed that the drug combination induces double-strand breaks and DNA fragmentation.

The main concern of this manuscript was to show the synergistic enhancement of the effects of MAPK inhibitors by Mcl-1 inhibition. Furthermore, we show that this enhancement is based on induction of apoptosis. Proving apoptosis induction, we have now collected the following data: DNA fragmentation, caspase activation, loss of MMP, AnnexinV/PI staining, now also PARP cleavage, cytochrome c release and induction of γH2AX.

All these data together clearly indicate induced apoptosis by S63845/SCH772984 treatment in melanoma cells.

Changes:

Abstract: The result of H2AX phosphorylation is mentioned.

Introduction: The relation of double strand breaks and DNA fragmentation is mentioned (§3).

Materials and methods: The used antibody for histone gH2AX is listed (2.6.Western blotting).

Results: The new gH2AX data are explained in 3.5.Significant role of caspase activation as well in Fig 7C. A new reference is added here (33. Rogakou et al., 2000).

Reviewer 4 Report

The manuscript by Peng et al. reported the combined inhibition of ERK and Mcl-1 by SCH772984 and S63845 resulted in enhanced apoptosis and loss of cell viability in melanoma cells. This work is of significant, and may provide a new strategy for overcoming drug resistance in melanoma therapy. However, the manuscript has some critical deficiency, making it unacceptable for publication in current form. If these issues could be addressed, it can be reconsidered for acceptance. The detailed comments are listed as following:

Major concerns:

1.     The drug combination index (CI) for BRAF or ERK1/2 inhibitors with Mcl-1 inhibitor should be provided as a figure or table.

2.     In Figure 4, the A375 cells treated with 1 µM S63845 alone did not induce cell death, which is inconsistent with results shown in Figure 2, please explain or discuss it.

3.     The results in Figure 8 show the pan-caspase inhibitor QVD-Oph can diminish loss of MMP and abolish ROS production induced by drug combination. However, ROS production and loss of MMP are often regarded as upstream stimulating effects of caspase activation, thus why QVD-Oph can block these effects, it is very confusing.

4.     Whether the ROS inhibitor (e.g. NAC) could block the combined inhibition of ERK and Mcl-1 in melanoma cells?

5.     Some bands of Western Blotting analysis show confusing results, such as caspase 8, 9 for MeWo cells, essentially unchanged. The authors should repeat these results or discuss the possible reason.

6.     The in vivo anti-tumor effect of drug combination for SCH772984 with S63845 is essential for this work. Thus, the authors should add the animal experiments to further demonstrate the conclusion.

Minor points:

1.     The figure legend title should be followed by the detailed figure legends. The font size of current form of figure legend seems very confusing.

2.     Page 3, line 11, other agonists” should be corrected as “inhibitors” or “agents”.

3.     The scale bar for immunofluorescence analysis is missing in Figure 6.

4.     The bands of WB results should be quantitated.

Author Response

Reviewer 4

Comments and Suggestions for Authors

The manuscript by Peng et al. reported the combined inhibition of ERK and Mcl-1 by SCH772984 and S63845 resulted in enhanced apoptosis and loss of cell viability in melanoma cells. This work is of significant, and may provide a new strategy for overcoming drug resistance in melanoma therapy. However, the manuscript has some critical deficiency, making it unacceptable for publication in current form. If these issues could be addressed, it can be reconsidered for acceptance. The detailed comments are listed as following:

Response: Thank you very much for the evaluation and the subsequent constructive and helpful comments. We addressed all of your comments. Please find the explanations below.

Major:

Comm 1.  The drug combination index (CI) for BRAF or ERK1/2 inhibitors with Mcl-1 inhibitor should be provided as a figure or table.

Response: Following this comment and in order to determine synergistic effects, the Web application SyngeryFinder 3.0 was applied [Ianevski et al., 2022]. The data of three concentrations of each drug were used. In particular, the Bliss scoring method was used, and delta scores of (δ) ≥ 10 were considered as synergistic, whereas scores between -10 and 10 were considered as additive. The calculations were done for the cell viability data (48 h, Fig 1), cell proliferation (24 h, 48 h, Fig 2) and apoptosis (48 h, Fig 3).

The calculations are presented in a graphical way, as suggested. As the manuscript has already 9 large figures, we suggest to present these data in a new supplementary figure (Fig S3). The calculated delta scores strongly suggest that the combination of SCH772984 and S63845 was synergistic in all 4 cell lines, as determined by three assays. On the other hand, the combination of vemurafenib/S63845 appeared as synergistic mainly in the two BRAF-mutated cell lines.

Changes:

Calculations are visualized in supplementary figure S3 and explained in the figure legend.

Score values are given and discussed in the Results text (3.3.Decreased cell viability correlates with induction of apoptosis).

Comm 2.     In Figure 4, the A375 cells treated with 1 µM S63845 alone did not induce cell death, which is inconsistent with results shown in Figure 2, please explain or discuss it.

Response:

To clarify this question, we repeated the AnnV/PI assay two more times. Now, the evaluation of four independent series of experiments (12 values) also showed some induction of cell death in A-375 in response to S63845 alone. Thus at 24 h, 4.9% +/- 2.1% AnnV-positive cells were found (Ctr: 2.2% +/- 0.2%, p < 0.002). At 48 h, 6.9% +/- 1.5% AnnV-positive cells were found (Ctr: 3.8% +/- 2.0%, p < 0.001). This effect was statistically significant, although weaker than the effect determined with the sub-G1 assay (14%, Fig 3).

We should consider that the results of Fig 3 and Fig 4, which were determined in two independent labs, are based on essentially different assays, which differ in sensitivity and reflect different phases in apoptosis. While sub-G1 (DNA fragmentation) is a late event, phospholipid flip-flop determined by Annexin V positivity represents an early step in apoptosis. The specific reason is not known, why A-375 cells respond less at the level of AnnV positivity. The major finding, however, that the proapoptotic effect of SCH772984 is strongly enhanced by the combination with S63845 is strongly supported by both assays.

Changes:

Fig 4 was substituted by the adjusted figure, now based on four independent experiments. Figure legend changed.

Re-calculated values are given in the Results text (3.3.Decreased cell viability correlates with induction of apoptosis)

R4, Comm 3.     The results in Figure 8 show the pan-caspase inhibitor QVD-Oph can diminish loss of MMP and abolish ROS production induced by drug combination. However, ROS production and loss of MMP are often regarded as upstream stimulating effects of caspase activation, thus why QVD-Oph can block these effects, it is very confusing.

Response:

Indeed, ROS production and loss of MMP are often upstream of caspase activation, as indicated. However, there is also several data suggesting the opposite direction under certain conditions.

Caspases often appear as upstream of the proapoptotic mitochondrial pathway, as caspase-8 can cleave and activate the proapoptotic Bcl-2 protein Bid, which then triggers loss of MMP.

BID cleavage may also be catalyzed by caspase-3, as shown in cytotoxic drug and UV radiation-induced apoptosis. This caspase-3-mediated Bid cleavage is considered as a positive feedback loop for amplification of apoptosis (Slee et al., 2000; Cell Death Differ. 7, 556–565).

Another example is apoptosis induced by p14ARF, which represents an alternative splice product to p16. Thus, p14ARF can trigger apoptosis via a caspase-3-dependent mitochondrial amplification loop (Milojkovic et al., 2013; Int J Cancer. 133:2551-62).

Another example is calicheamicin-induced apoptosis, where caspase inhibition by zVAD was shown to interfere with mitochondrial activation. This again places caspase activation upstream of the mitochondria and indicates an amplification loop that leads from caspase activation to the mitochondrial pathway (Prokop et al., 2003; Oncogene. 22:9107-20).

The relationship between caspase activation and ROS is still largely unclear and remains controversial. However, in several models, investigators reported that the generation of ROS can also occur downstream of mitochondrial activation (Cai et al., 1998; Biol. Chem. 273:11401-11404; Gottlieb, et al., 2000; Mol. Cell. Biol. 20:5680-5689, Kirkland et al., 2002; J. Neurosci 22:6480-6490).

We had previously shown that the pancaspase inhibitor zVAD can prevent ROS production in melanoma cells, e.g. when ROS were induced by the iron-containing cytosine analogue N69, indicating that ROS production may come downstream of proteases (Franke et al., 2010; Biochem Pharmacol 79:575-86).

Interestingly, complex I and II of the electron transport chain had been identified as early caspase targets upon their activation via the mitochondrial pathway. Disruption of their function by caspases can result in mitochondrial damage, loss of MMP and ROS generation (Ricci, et al., 2003; J. Cell Biol. 160:65–75; Ricci, et al., 2004; Cell. 117:773-786).

In the present manuscript, we show that ROS production in course of SCH772984/S63845 treatment comes relatively late (24 h), which also suggests ROS as a secondary effect. The exact role of ROS in the complex interplay of signaling pathways remains to be elucidated.

Changes:

Discussion: These interrelations of the pathways are briefly addressed (§13).

Rev 2, Comm 4.     Whether the ROS inhibitor (e.g. NAC) could block the combined inhibition of ERK and Mcl-1 in melanoma cells?

Response: We have some experience with the use of NAC and other antioxidants from previous projects. Thus, the indirubin derivative DKP-073 resulted in characteristic and early ROS production in melanoma cells. In this setting, NAC (800 µM) completely abrogated ROS production as well as it completely prevented DKP-073-induced apoptosis (Zhivkova et al., 2019; Mol Carcinog 58:258-269, doi: 10.1002/mc.22924).

Thus, we also tried to prevent ROS production in course of SCH772984/S63845 treatment by using NAC (1 mM), as suggested. As an alternative antioxidative strategy, we also tried to suppress ROS by tocopherol (vitamin E, 1 mM). However, ROS production in course of SCH772984/S63845 treatment could not be prevented by antioxidants, neither by NAC nor by tocopherol (new Fig S6A). In consequence, NAC and tocopherol had no significant effect on SCH772984/S63845-induced loss of cell viability (new Fig S5B) and on induced apoptosis (new Fig S6C). To check the activity of NAC, we used DKP-073 (from the previous project) again as control. Proving the full antioxidative activity of NAC in these experiments, DKP-073-induced ROS production was completely prevented by NAC (Fig S6A).

There are different types of ROS, which are further located at different cellular compartments. Thus, the antioxidative capacity of typical antioxidants as NAC and tocopherol sometimes is not sufficient. We found this also in previous projects, e.g. when melanoma cells were treated with the iron-containing cytosine analogue N69 (Franke et al., 2010; Biochem Pharmacol 79:575-86). We do not think that ROS played the decisive role in the present setting, as ROS was induced only later at 24 h after treatment. For other ROS-related strategies, we reported very early ROS production (< 4 h), e.g. for indiribin and N69, cited above.

Changes:

The use of NAC and tocopherol is described in materials and methods.

The findings are explained in results (3.5.Significant role of caspase activation, last paragraph) and are shown in the new Fig S6.

The negative findings on ROS production at earlier time (4 h) is mentioned as data not shown. 2.4.Determination of mitochondrial membrane potential and reactive oxygen species (ROS).

The role of ROS is briefly addressed in the discussion (§13). A new reference is added here (66 Franke et al., 2010).

Rev 2, Comm 5.     Some bands of Western Blotting analysis show confusing results, such as caspase 8, 9 for MeWo cells, essentially unchanged. The authors should repeat these results or discuss the possible reason.

Response: As for caspase-3, we had previously used a cleaved caspase-3 antibody; which indeed showed only weak bands for MeWo (15, 17 kDa; now Fig 7B). As suggested, we repeated the experiments with an alternative total caspase-3 antibody, which now also showed clear activation of caspase 3 in MeWo (Fig 7C).

As for caspase-9, the previous antibody only detected cleavage products at 37 kDa. Thus, we repeated the caspase-9 WB by applying another caspase-9 antibody. This new antibody now also detected the 20 kDa final processing product of caspase-9. The 20 kDa product showed a clear induction also in MeWo.

Thus, the use of the additional antibodies confirmed that caspases were activated by SCH772984/S63845 also in MeWo cells. The intensity of caspase cleavage products not necessarily must reflect the real caspase activity in cells as also degradation and the half-life of cleavage products contribute. We thus analyzed as an additional assay the processing of PARP, a well-known target of caspase-3. Clear processing of PARP was seen both in MeWo and A-375 cells (Fig 7C). To further address the question of caspase involvement, we performed caspase activity assays (FLICA). These quantitative assays proved that caspase activity was induced in 37% of MeWo cells and in 28% of A-375 (Fig S5).

These data together clearly underline the involvement of caspases in both investigated melanoma cell lines in course of treatment with SCH772984/S63845.

Changes:

Materials and methods: New antibodies are listed (2.6.Western blotting). The caspase activity assays are described (2.5. Caspase activation assay and cytochrome c release).

Results: New data of total caspase-3, caspase-9 and PARP antibodies as well as results of FLICA caspase activity assays are described (3.5. Significant role of caspase activation).

Figures: New data are illustrated in an expanded figure 7 as well as in in a new supplementary Fig S5.

Rev 2, Comm 6.     The in vivo anti-tumor effect of drug combination for SCH772984 with S63845 is essential for this work. Thus, the authors should add the animal experiments to further demonstrate the conclusion.

Response: We would like to recall the general concerns of our manuscript. Although vemurafenib and other BRAF inhibitors have led to a breakthrough in melanoma therapy, still many patients die due to induced BRAFi resistance. We are thus happy that we can suggest here a strategy, which strongly increases the efficacy of BRAF inhibitors and thus may overcome or decrease the resistance problems. Furthermore, this study underlines that ERK inhibitors may provide an alternative strategy, which may apply also for BRAF-WT melanomas. Again, the efficacy of ERK inhibition is strongly enhanced by the used Mcl-1 inhibitor. This information may be helpful in future, possibly to decide whether ERK inhibitors may be of use in melanoma therapy. Increased effectiveness may allow to reduce the drug dose, which may also reduce the side effects.

The regulation of apoptosis in melanoma cells is still an important question. Here, we present a comprehensive series of experiments, which shed more light on this issue. Even more experiments have been added in this revision in response to the comments of five reviewers.

Thus, we think that the information given is this manuscript is already extensive and also important. We further think that it is not really justified at this early point to ask for additional animal experiments. It is clear that animal experiments have a long start-up time. We may need 3 months alone to get a corresponding application through, and then another 3 months for performing the animal experiments. The next question is, whether we may already get the right answer with a single set of animal experiments. Rather, it may be necessary to solve the questions of dosing, drug application, half-life, turnover and reproducibility in previous pilot and/or additional experiments.

On the other hand it is important to note that the use of Mcl-1 inhibitors and ERK inhibitors in melanoma is presently of high actuality, and in 6-12 months, there will be several other papers out with related or even overlapping results.

To address the reviewer's comment at least in part and in a timely manner, we determined the effects of the drug combination on apoptosis and cell viability in three cultures of normal human fibroblasts (ATCC). The fibroblast cultures responded much less to the combination of S63845 / SCH772984 (each 1 µM) as compared to the melanoma cells, revealing a reduced cell viability of only 82% +/- 7% and an induction of apoptosis of only 11% +/-2%. As there are already nine figures in the manuscript, we suggest to show the fibroblast data in an additional supplementary figure (new Fig S4).

It is true that the drug combinations suggested in our manuscript were not tested so far in vivo. On the other hand, the single treatments are out of question. Of course, vemurafenib and other BRAF inhibitors are used in the clinic since many years. The selective ERK1/2 inhibitor SCH772984 has been tested in vivo in xenograft melanoma models [Samatar et al., 2014; Nat Rev Drug Discov 13: 928-942, doi:10.1038/nrd4281]. Other ERK inhibitors as Ulixertinib and MK-8353 have already been tested in phase 1 clinical trials [Moschos et al., 2018; JCI Insight 3, doi:10.1172/jci.insight.92352; Varga et al., 2020; Clin Cancer Res 26: 1229-1236, doi:10.1158/1078-0432.CCR-19-2574]. As for Mcl-1 inhibitors, four drugs (S64315, AMG-176, AMG-397, and AZD-5991) have already entered clinical evaluation in patients with hematological malignancies [Senichkin et al., 2020; Cell Death Differ 27: 405-419]. Of these, S64315 (MIK665) is chemically related to the here used S63845, and shows comparable activity [Szlávik et al., 2019; J Med Chem 62: 6913–6924].

Thus, there seems to be good hope that also the combination of these drugs may be tolerated in patients, in particular as the here shown mutual enhancement may allow to decrease the single doses.

Changes:

These issues are now better discussed (Discussion, §11). One additional reference is added here (61. Szlávik et al., 2019).

Normal human fibroblasts cultures are described in materials and methods (2.1.Cell culture and treatment).

Response of fibroblasts to S63845 / S63845 is explained in Results section “3.3. Decreased cell viability correlates with induction of apoptosis” as well as in the supplementary Fig S4.

Minor points:

  1. The figure legend title should be followed by the detailed figure legends. The font size of current form of figure legend seems very confusing.

Response: We see the format problem. The setting had been done by the publisher. We now changed it for all figure legends.

  1. Now the legends directly follow the titles
  2. The fond size of the text fields was adapted to that of the title (Palantino 9), while the main text is in Palantino 10.
  3. The figure titles were formatted in bold
  4. All legends were formatted in justification

  1. Page 3, line 11, “other agonists” should be corrected as “inhibitors” or “agents”.

Response: has been changed to “agents”.

  1. The scale bar for immunofluorescence analysis is missing in Figure 6.

Response: Scale bars have been inserted in figure 6.

Changes: Fig 6 and legend

Rev 3, comm 4.     The bands of WB results should be quantitated.

Response: We had quantified already the WB data of all Bcl-2 proteins as well as the respective GAPDH controls. Thus, median induction factors were calculated on basis of densitometric, semiquantitative analyses and normalization with the GAPDH signals from two independent series of protein extracts and Western blots. Upregulation of Mcl-1, BimEL, Noxa and Puma is indicated as fold increase, while downregulation of Bcl-2 and pBad is indicated as percent in Results, 3.6. Upregulation of proapoptotic BH3-only proteins. The other Bcl-2 proteins did not show a significant regulation.

Further following the suggestion of the reviewer, we have now additionally quantified the downregulation of phosphorylated ERK as well as some new data suggested by reviewer 3: PARP cleavage, phosphorylated histone H2AX (gH2AX) and mitochondrial cytochrome c release. As activated caspases were absent in controls and in cells treated with S63845 or SCH772984 alone, a quantification was not reliable. PARP, gH2AX and cytosolic cytochrome C were also not seen in controls and SCH772984-treated cells. Here we compared combination-treated cells with the samples treated with S63845 alone and calculated respective induction factors.

Changes: The quantification of Western blots is explained in Materials and methods, 2.7. Statistical analyses.

The quantitative data is given in results, section 3.5. Significant role of caspase activation as well as in 3.6. Upregulation of proapoptotic BH3-only proteins.

All quantification data is supplied in the “Original Western blots” file

Reviewer 5 Report

To authors

The present manuscript by Peng et. al deals with the effects of MAPK inhibitors in combination with Mcl-1 inhibitors on cell viability and apoptosis in melanoma.  These effects were assessed by several methods including western blots, WST-1 assay, flow cytometry, fluorescent microscopy analysis. Authors concluded that combined treatment of ERK and Mcl-1 inhibitors showed an impressive efficiency both in BRAF-WT and -mutated melanoma.  This is a very well-designed, well-controlled study, and well-written.

Minor points:

・      In figure 7, pERK (24) <- you need to add a character “h” after 24

・      In figure 7, Csp-3 (24 h) <- you need to add a word “cleaved” before Csp-3 because this antibody recognizes cleaved Csp-3 not its total protein

Author Response

Reviewer 5

Comments and Suggestions for Authors

The present manuscript by Peng et. al deals with the effects of MAPK inhibitors in combination with Mcl-1 inhibitors on cell viability and apoptosis in melanoma.  These effects were assessed by several methods including western blots, WST-1 assay, flow cytometry, fluorescent microscopy analysis. Authors concluded that combined treatment of ERK and Mcl-1 inhibitors showed an impressive efficiency both in BRAF-WT and -mutated melanoma.  This is a very well-designed, well-controlled study, and well-written.

Response: Thank you very much for the positive evaluation.

Minor points:

・      In figure 7, pERK (24) <- you need to add a character “h” after 24

・      In figure 7, Csp-3 (24 h) <- you need to add a word “cleaved” before Csp-3 because this antibody recognizes cleaved Csp-3 not its total protein

Response: Thank you, both were changed. Due to comments of other reviewers, figure 7 was re-organized.

Round 2

Reviewer 3 Report

The author has answered almost all of my queries. The manuscript can be accepted in present form.

Reviewer 4 Report

The authors have addressed most of my comments and this paper has greatly improved. Thus the paper can be accepted in present form.